# Systematic benchmarking of tools for CpG methylation detection from nanopore sequencing

Zaka Wing-Sze Yuen [1,2], Akanksha Srivastava [1,2], Runa Daniel[3], Dennis McNevin [4], Cameron Jack [2✉] & Eduardo Eyras [1,2,5,6✉]

DNA methylation plays a fundamental role in the control of gene expression and genome integrity. Although there are multiple tools that enable its detection from Nanopore sequencing, their accuracy remains largely unknown. Here, we present a systematic benchmarking of tools for the detection of CpG methylation from Nanopore sequencing using individual reads, control mixtures of methylated and unmethylated reads, and bisulfite sequencing. We found that tools have a tradeoff between false positives and false negatives and present a high dispersion with respect to the expected methylation frequency values. We described various strategies to improve the accuracy of these tools, including a consensus approach, METEORE (https://github.com/comprna/METEORE), based on the combination of the predictions from two or more tools that shows improved accuracy over individual tools. Snakemake pipelines are also provided for reproducibility and to enable the systematic application of our analyses to other datasets.

[1] EMBL Australia Partner Laboratory Network, Australian National University, Canberra, ACT, Australia. [2] The John Curtin School of Medical Research, Australian National University, Canberra, ACT, Australia. [3] Office of the Chief Forensic Scientist, Victoria Police Forensic Services Department, Macleod, VIC, Australia. [4] Centre for Forensic Science, School of Mathematical & Physical Sciences (MaPS), Faculty of Science, University of Technology Sydney, Sydney, NSW, Australia. [5] Catalan Institution for Research and Advanced Studies (ICREA), Barcelona, Spain. [6] Hospital del Mar Medical Research Institute (IMIM), Barcelona, Spain. ✉email: cameron.jack@anu.edu.au; eduardo.eyras@anu.edu.au

DNA modifications play a fundamental role in genome stability and gene regulation during mammalian development, disease progression, and aging[1–4]. Of more than 17 possible DNA modifications, the methylation of cytosines at CG di-nucleotides (CpG), involving the addition of a methyl group (−CH₃) to the 5th carbon of the cytosine ring to form 5-methylcytosine (5mC), is the most frequently observed methylation in relation to gene regulation[5]. Key advances in the understanding of the function of 5mC have been made possible through the development of dedicated genome-wide profiling techniques[4]. Commonly used methods include restriction enzyme digestion, affinity enrichment, or bisulfite conversion, followed by microarray hybridization or short-read sequencing[4]. While short-read sequencing has been very effective at mapping 5mC sites at genome-scale, it still presents various disadvantages including high mapping uncertainty in repetitive regions and amplification bias. Moreover, short-read approaches result in the loss of native biochemical modifications and involve the necessary coupling with lengthy protocols, such as a bisulfite conversion, which is known to degrade DNA[6]. The conversion of 5mC to uracil is also very sensitive to the reaction conditions[7].

In contrast, nanopore long-read technologies provide many distinct advantages. Individual DNA molecules, harboring base modifications, can be sequenced in their native state without any prior enzymatic or chemical treatment and without the need for PCR amplification[8,9]. As a single-DNA molecule travels through a pore, base modifications can be revealed by their unique signal shapes, which differ from the equivalent unmodified base[8–10]. However, signals are dependent on sequence context and different copies of the same molecule present considerable signal variation. Thus, it is necessary to apply computational models to interpret the signals and to predict the methylation status at a given CpG site. Multiple tools have been developed in recent years, but a lack of systematic benchmarking poses a significant challenge for users in assessing the reliability of their predictions. Although nanopore provides an opportunity to detect methylation at the single-molecule level, its accuracy in this context remains largely unknown. This precludes the development of reliable and cost-effective applications in medical, forensic, and environmental samples. It is thus necessary to thoroughly characterize the strengths and limitations of the different available tools and establish different strategies for the accurate detection of 5mC in DNA.

We have performed a systematic benchmarking of six tools for 5mC detection from nanopore sequencing fast5 files using individual reads, controlled methylation mixtures, Cas9-targeted sequencing, and whole-genome bisulfite sequencing (WGBS). The detection capabilities of the six tools were established at the single-molecule level and in discerning different stoichiometries, at different levels of coverage and GC content. In general, the tested tools present a trade-off between true positives and false positives, and a high dispersion in the prediction of methylation frequencies. We propose various strategies to improve detection accuracy, including a consensus approach, METEORE (https://github.com/comprna/METEORE)[11], that combines the outputs from two or more tools.

## Results

**Detection of DNA cytosine methylation from nanopore sequencing.** A standardized workflow was developed to obtain 5mC calls at CpG sites from Nanopolish[8], Megalodon[12], DeepSignal[13], Guppy[14], Tombo[15], and DeepMod[16] (Fig. 1), using Snakemake

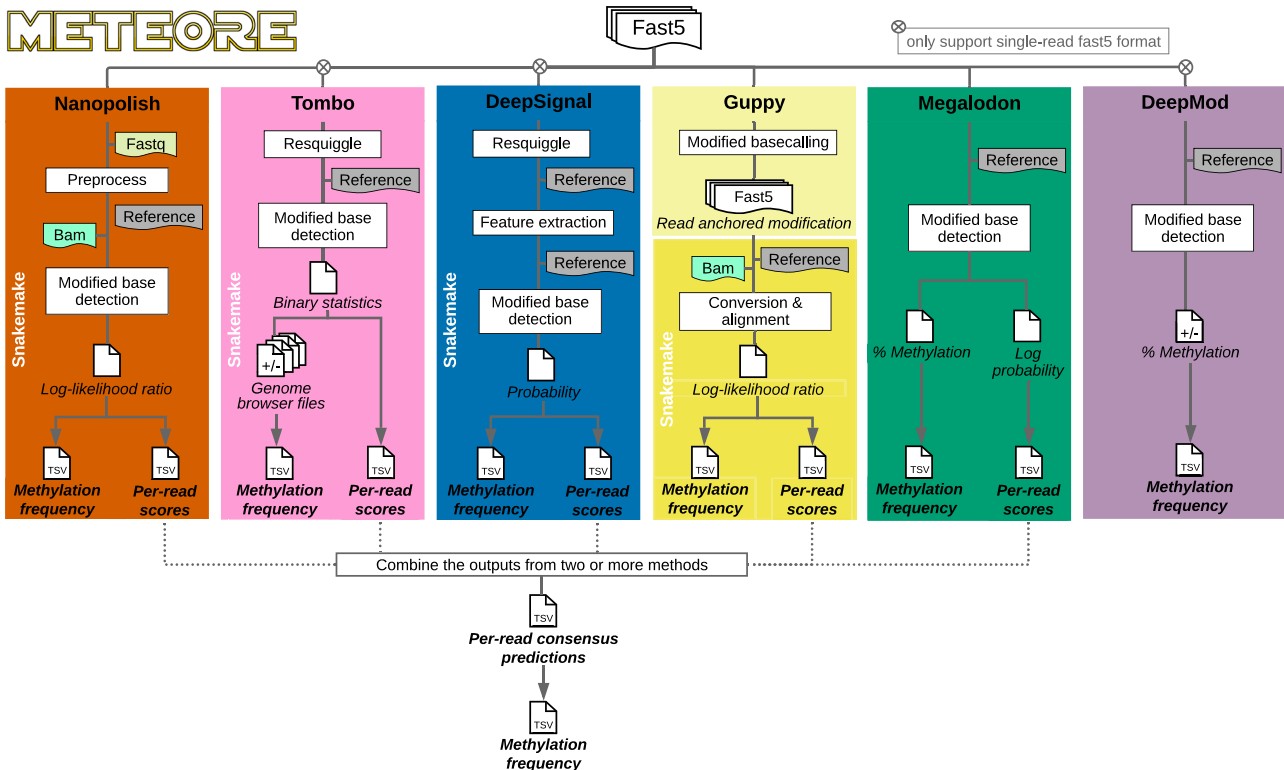

**Fig. 1 Analysis pipeline for 5mC detection at CpG sites from nanopore sequencing.** The diagram describes the approach used to test the tools Nanopolish, Tombo, DeepSignal, Megalodon, Guppy, and DeepMod. Snakemake pipelines and command lines used are available at https://github.com/comprna/METEORE[11]. A Snakemake pipeline was not developed for Megalodon and DeepMod as they can be run with a single command. Excluding DeepMod, all tools produce predictions per individual read and per CG site. In addition, all tools predict the methylation frequency at each genome site from fast5 input files. Methods that currently only accept single-read fast5 format are indicated.

pipelines[17] (https://github.com/comprna/METEORE)[11]. This workflow ensures consistent inputs and outputs for all tools and facilitates the integration and interpretation of DNA methylation calls. Nanopolish detects CpG methylation with a hidden Markov model, while Megalodon, DeepSignal, and DeepMod use neural networks, and Tombo applies a statistical test to identify DNA modifications. Both Tombo and DeepSignal resquiggle the raw signals before detection. Differently from the other tools, Guppy directly basecalls 5mC using an extended alphabet. Megalodon anchors the Guppy basecalling output to the reference and assigns a score for the candidate modified base. All tools output per-site methylation calls on both strands at genome level.

**Performance of CpG methylation detection on methylation controls.** We first evaluated the six tools on methylation control datasets built from PCR-amplified DNA (negative control) and M.SssI-treated DNA (positive control)[8]. From the 346,793 CpG sites in the *E. coli* reference genome, we selected 100 arbitrary sites (Supplementary Data 1, Supplementary Fig. 1a) containing a single CpG site with a 10 nt window on either side with no CGs, and with a minimum read coverage of 50× (median coverage 85×), from both positive (methylated) and negative (unmethylated) control datasets. Using these reads, we created 11 benchmarking datasets with specific mixtures of methylated and unmethylated reads, namely, containing 0%, 10%, …, 90%, and 100% of methylated reads, each set with ~2400 reads (mixture dataset 1) (Supplementary Table 1, Supplementary Data 3). We examined the per-site methylation frequency predicted by each tool across these 100 sites using the default cutoff for each method (Supplementary Data 5). All tools, except Tombo, achieved Pearson correlations above 0.8 (*p* value < 2.2e−16 for all tools) (Fig. 2a, Supplementary Fig. 2). The highest correlation and lowest root mean square error (RMSE) values were attained by Megalodon, followed by DeepMod and DeepSignal.

Despite the good correlation values, most tools showed high dispersion and low agreement with the expected percentage methylation per site. Guppy had the highest RMSE values and systematically underpredicted the per-site methylation (Fig. 2a). Nanopolish and Tombo showed high dispersion and systematically overpredicted. To further assess how the dispersion affected the per-site accuracy, we calculated the proportion of sites predicted outside a 10% window around the expected value for each percentage methylation subset. Guppy had most sites predicted outside expected windows (Fig. 2b). Nanopolish and Tombo had the lowest proportion of sites predicted outside the m90 and m100 windows, but the highest proportions at low methylation frequency (Fig. 2b). In contrast, Megalodon had most sites predicted within the expected windows in low methylation subsets (Fig. 2b).

We further assessed the classification of fully unmethylated and fully methylated sites according to specific thresholds (Supplementary Table 2, Supplementary Data 6 and 7). For 0% methylation, Guppy and Megalodon correctly recovered all sites as unmethylated if a methylation proportion < 0.1 was considered unmethylated, whereas other methods predicted correctly fewer unmethylated sites (Fig. 2c). In the 100% methylated set, using a methylation proportion > 0.8, only Nanopolish, Tombo, and Megalodon correctly predicted most of the sites as fully methylated (Fig. 2d). In contrast, Guppy failed to predict any sites at this cutoff (Fig. 2d). These differences in the accuracy at fully methylated and fully unmethylated sites, as well as the general high dispersion observed at intermediate methylation levels, motivated us to identify alternative strategies to achieve higher accuracies.

**Methylation prediction accuracy in individual molecules.** Nanopore sequencing provides the opportunity to detect nucleotides and their modifications in individual molecules. We therefore explored the accuracy of the tools for identifying 5mC sites in individual reads. The per-read performance of each tool was evaluated across their range of prediction scores at each site in individual reads (Supplementary Data 8 and 9), excluding DeepMod which only provides the percentage of methylation per site, so could not be included in this analysis. All tested tools achieved areas under the receiver operating characteristic (ROC) curve (AUC) (Fig. 3a) and areas under the precision–recall (PR) curve AUC_PR above 0.8 (Fig. 3b). Megalodon showed the highest AUC and AUC_PR values, followed by DeepSignal and Nanopolish (Fig. 3b). Guppy had decreased precision at high recall values (Fig. 3b). The differing accuracies of methods across different conditions suggested that a consensus approach may capture the advantages of the methods and compensate for their potential deficiencies.

**Combination of predictions in individual molecules improve accuracy.** We developed a consensus approach, METEORE[11], that combines the predictions from two or more tools. The consensus was implemented using two different models, a random forest (RF) and a multiple linear regression (REG). The combination of two methods using either of these two models provided overall an increase in accuracy compared to individual methods (Fig. 3c, d, Supplementary Fig. 3). The five individual tools were then compared with METEORE (RF and REG) for the combination of Megalodon and DeepSignal, on a different collection of methylation mixtures (mixture dataset 2) (Supplementary Table 1, Supplementary Data 2 and 4). The predictions were performed on individual reads and then summarized per CG site to compare with the expected percentage methylation (Supplementary Data 10). METEORE RF combining Megalodon and DeepSignal achieved lower RMSE compared with the individual tools (Fig. 3e). METEORE REG performed similarly to Megalodon, and improved accuracy over the other tools (Fig. 3e). METEORE RF also showed an improvement in the proportion of sites predicted within the expected 10% window at intermediate and high methylation sets (Fig. 3f).

**Varying single score cutoffs improve the accuracy of methylation predictions.** At this stage, the tools were applied with their default score cutoffs. We reasoned that it might be possible to identify different cutoffs for the scores at the individual read level to improve the accuracy of the predictions of methylation frequency per site. To do this, we considered the distribution of scores (Supplementary Fig. 4a) and several accuracy metrics using individual reads from the mixture dataset 1 (Supplementary Fig. 4b). We then determined for each method the score that corresponded to the maximum of true positive rate (TPR)–false-positive rate (FPR) (Supplementary Table 3). Applying these scores to separate methylated and unmethylated sites per read on the mixture dataset 2, all tools had a lower RMSE value with respect to the default cutoffs (Supplementary Fig. 5a). In terms of correlation, all tools except Nanopolish and Megalodon improved. Selecting the score corresponding to the minimum of $FPR^2 + (1 - TPR)^2$ led to similar cutoffs (Supplementary Table 3) and results (Supplementary Fig. 5b). Applying these cutoffs to METEORE (RF and REG) combining Megalodon and DeepSignal improved upon the default cutoffs and achieved a lower RMSE compared with the individual tools (Supplementary Fig. 5a, b, Fig. 3e). Nevertheless, all tools still showed a high proportion of predicted sites outside the expected 10% window. In particular, Nanopolish and Megalodon had more sites

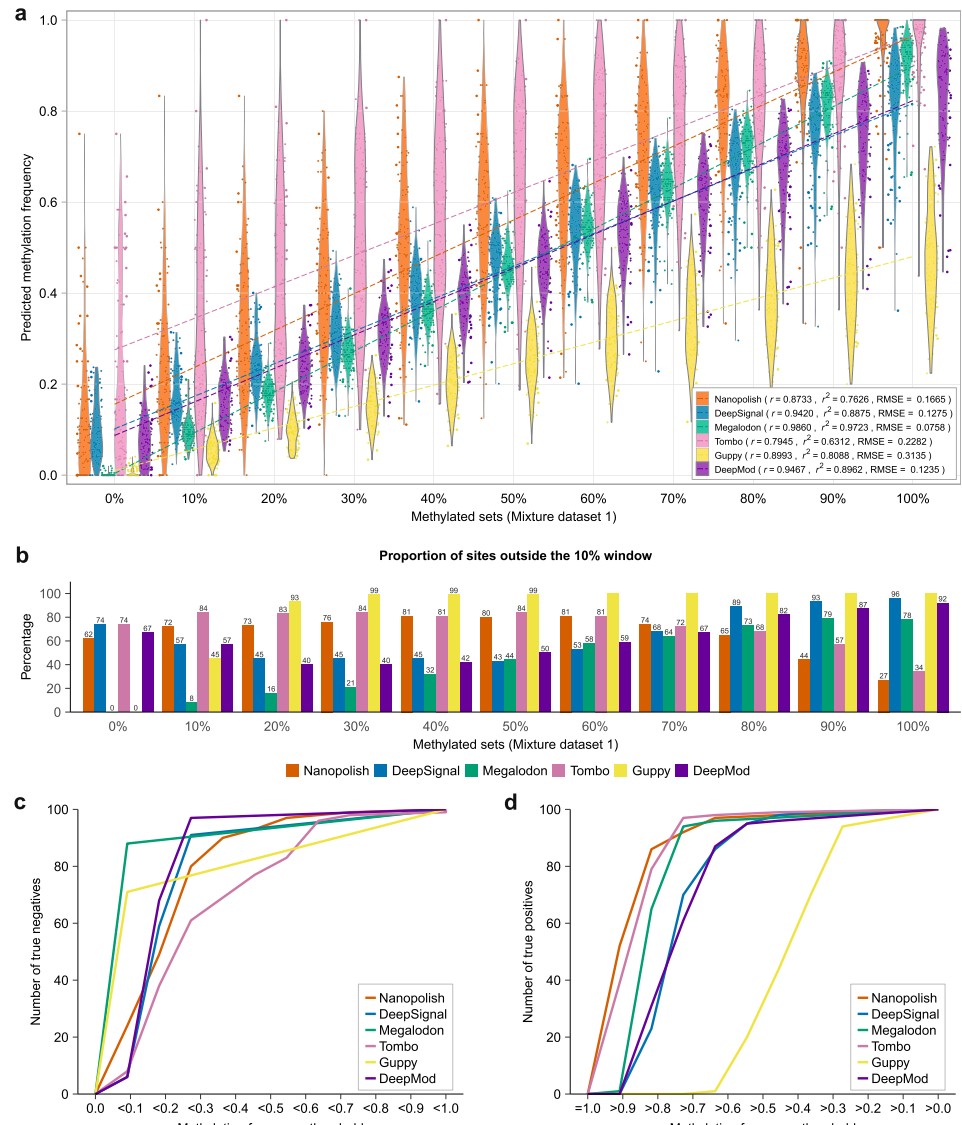

**Fig. 2 Accuracy analysis per CpG site on control mixture dataset 1. a** Violin plots showing the predicted methylation frequencies (*y* axis) for each control mixture set with a given proportion of methylated reads (*x* axis). The Pearson's correlation (*r*), coefficient of determination (*r²*), and root mean square error (RMSE) are given for each tool. **b** For each method, we indicate the proportion of sites predicted outside a 10% window around the expected methylation proportion, i.e., each predicted site in the m% dataset was classified as "outside" if its predicted percentage methylation was outside the interval [(m − 5)%, (m + 5)%] for intermediate methylation values, or outside the intervals [0,5%] or [95%, 100%] for the fully unmethylated or fully methylated sets, respectively. The percentage is indicated on top of each bar, except for 100%. **c** Empirical cumulative distribution function (ECDF) plot showing the number of true negatives (*y* axis) for each tool according to different thresholds for the predicted methylation frequency below which a site was called unmethylated (*x* axis), using the dataset of 100 fully unmethylated sites. **d** ECDF plot showing the number of true positives (*y* axis) for each tool according to different thresholds for the predicted methylation frequency above which a site was called fully methylated (*x* axis), using the dataset of 100 fully methylated sites.

predicted outside the expected windows at high and low methylation sets, respectively (Supplementary Fig. 5c, d). In contrast, METEORE RF and REG had fewer sites predicted outside the windows at low methylation and high methylation respectively (Supplementary Fig. 5c, d).

**Discarding reads of uncertain methylation prediction state improves accuracy.** We considered the alternative strategy of discarding sites in reads with uncertain methylation state. This is used by default in Nanopolish and Tombo, which define a double cutoff (higher and lower than the point of indecision). To test this strategy, we used the distribution of scores (Supplementary Fig. 6)

and removed the 10% of sites in individual reads that were closest to the score at which the FPR and 1-TPR curves crossed (Supplementary Table 4). Using this approach, all methods except Megalodon achieved higher correlation and lower RMSE values compared with the default cutoffs (Supplementary Fig. 6a). We also assessed the scores at which FPR = 0.05 and 1-TPR = 0.05 and removed all predictions between these two values (Supplementary Table 4). This led to improved performance of all methods with respect to default cutoffs, except for Megalodon and Nanopolish (Supplementary Fig. 6b). In addition, we observed that Nanopolish used much fewer reads compared to the other methods (Supplementary Fig. 6c).

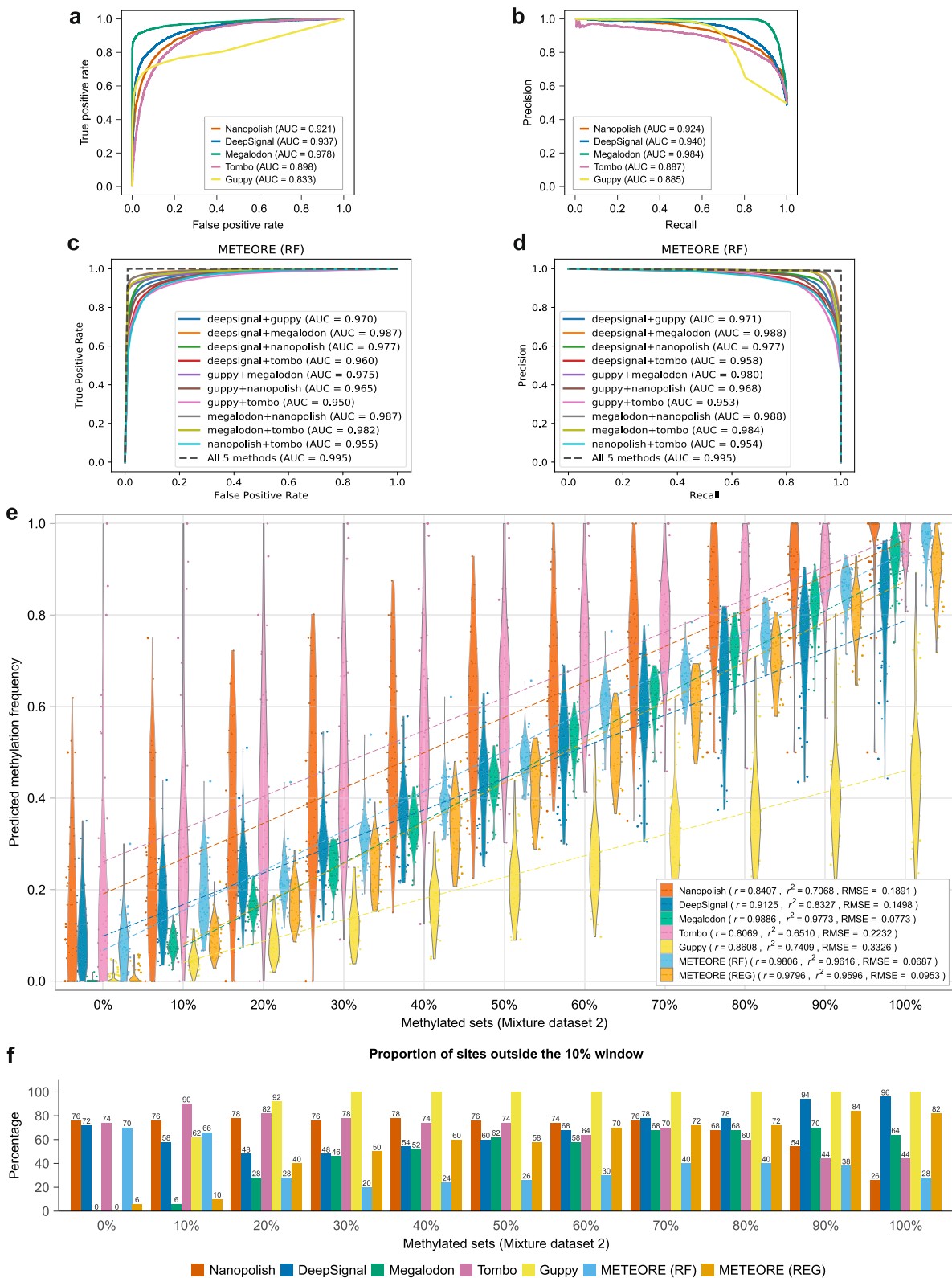

**Nanopore recapitulates bisulfite sequencing data at sites of low and high methylation**. We next performed a comparison with WGBS, which is one of the most used techniques to study CpG methylation at a genome-wide scale. To achieve enough read coverage, we used the Cas9-targeted nanopore sequencing (nCATS) protocol[18]. We selected ten regions of forensic relevance to sequence the native nuclear DNA of a human lymphoblastoid

cell line (NA12878) with a MinION flowcell (Supplementary Table 5, Supplementary Fig. 7). We used the reads corresponding to the targeted regions to analyze the CpG methylation patterns with the tested tools using default cutoffs, and compared the results with existing WGBS data for NA12878 from the Encyclopedia of DNA Elements (ENCODE) project[19] (Supplementary Data 11). Combining the methylation predictions from both on

**Fig. 3 Model accuracy at the individual read level and per-site accuracy analysis on control mixture dataset 2. a** Receiver operating characteristic (ROC) curves showing the false-positive rate (*x* axis) and true positive rate (*y* axis) for the predictions at individual read levels for the five methods tested, using reads from 0 and 100% methylated sets. **b** Precision–recall (PR) curves showing the recall (*x* axis) and precision (*y* axis) for the predictions at individual read levels for the five methods tested, using reads from 0 and 100% methylated sets. **c** ROC curves for METEORE for the random forest (RF) model (parameters: max_depth = 3 and n_estimator = 10) combining two methods, as well as combining the five methods. The curves were built from the average of a tenfold cross-validation with mixture dataset 1. Similar plots for another RF model using default parameters and a regression (REG) model are shown in Supplementary Fig. 3. **d** PR curves for the same models as in **c**. **e** Violin plots showing the predicted methylation frequencies (*y* axis) for each control mixture set with a given proportion of methylated reads (*x* axis) from the mixture dataset 2 for the five tested tools plus METEORE combining Megalodon and DeepSignal using a random forest (RF) or a regression (REG) model. The Pearson's correlation (*r*) and coefficient of determination (*r*$^2$) are given for each tool. **f** We indicate the proportion of sites predicted outside a window around the expected methylation proportion, i.e., a site in the m% dataset was "outside" if the predicted percentage methylation was outside the interval [(m − 5)%, (m + 5)%] for intermediate methylation sets, or outside the intervals [0,5%] or [95%,100%] for the fully unmethylated or fully methylated sets, respectively. The percentage is indicated on top of each bar, except for 100%.

**Table 1 Comparison of CpG methylation frequencies from whole-genome bisulfite sequencing (WGBS) and from Cas9-targeted nanopore data.**

|               | N    | r      | r$^2$   | ρ      | RMSE   |
|---------------|------|--------|--------|--------|--------|
| Nanopolish    | 1704 | 0.8652 | 0.7485 | 0.8326 | 0.2248 |
| DeepSignal    | 1731 | 0.9177 | 0.8423 | 0.8765 | 0.1708 |
| Megalodon     | 1723 | 0.9117 | 0.8312 | 0.8801 | 0.1772 |
| Tombo         | 1661 | 0.7765 | 0.6030 | 0.7537 | 0.2996 |
| Guppy         | 1738 | 0.8513 | 0.7246 | 0.8316 | 0.2334 |
| DeepMod       | 1739 | 0.7401 | 0.5477 | 0.7264 | 0.2874 |
| METEORE (RF)  | 1723 | 0.9174 | 0.8416 | 0.8862 | 0.1829 |
| METEORE (REG) | 1723 | 0.9262 | 0.8579 | 0.8885 | 0.1607 |

For each method, we give the number of sites (*N*), the Pearson's correlation (*r*), coefficient of determination (*r*$^2$), the Spearman's rank correlation (ρ), and the root mean square error (RMSE) for the comparison of the percentage methylation predicted from nanopore with the percentage methylation calculated from WGBS Illumina data. We show the results for the six tested tools and METEORE combining Megalodon and DeepSignal using a random forest (RF) (parameters: max_depth = 3 and n_estimator = 10) or a regression (REG) model.

CpG sites showed an improved correlation for all tools compared with using the methylation prediction independently for each strand (Supplementary Fig. 8). Using the combined methylation from both strands, all tools showed a positive correlation with the WGBS signals (Table 1). METEORE (REG) combining Megalodon and DeepSignal achieved the highest Pearson correlation and lowest RMSE values (Table 1).

To compare the spread of methylation predictions, we categorized the WGBS data into three bins of increasing methylation frequency (Fig. 4a). Tombo showed the largest dispersion of values at low (0.0–0.3) and intermediate (0.3–0.7) methylation, Guppy and DeepMod underpredicted at high methylation (0.7–1.0), and all methods overpredicted at intermediate methylation (Fig. 4a). The same analyses on similar experimental datasets from eight other regions from Gilpatrick et al.[18] (Supplementary Fig. 9, Supplementary Data 12) showed similar trends (Supplementary Figs. 10 and 11). For this dataset, all methods achieved higher correlations and lower RMSE values with WGBS data (Supplementary Table 7), possibly because these regions contained more CpG sites with high or low methylation[18].

**Methylation accuracy is stable at low coverage.** Overall, correlations between the nanopore methylation data and WGBS were approximately constant for all tools across different levels of coverage (Fig. 4b, Supplementary Fig. 12a). Megalodon, Deep-Signal, and METEORE had correlations above 0.9, whereas Tombo had lower correlations across all coverage levels, which agreed with our findings that Tombo was less accurate at separating fully methylated from fully unmethylated sites (Fig. 4b,

Supplementary Fig. 12a). Moreover, both Nanopolish and Tombo generally reported much lower mean coverage of reads compared with other tools, consistent with their application of double cut-offs to discard reads (Fig. 4c, Supplementary Fig 12b). The same analyses with data from Gilpatrick et al.[18] also showed stable correlations with WGBS at most levels of coverage for all tools, with a marked drop at high minimum coverage (Supplementary Fig. 11b, c). The general drop in accuracy at high minimum input coverage was due to the reduced number of sites available. To compensate for this, we subsampled different coverage levels (5×, 10×, 20×, 50×) using a fixed number of sites, and found that the correlation increased slightly with increased coverage (Supplementary Table 8).

**Nanopore recovers the methylation patterns along genomic regions.** We compared the methylation profiles from WGBS and nanopore along our ten tested regions (Fig. 5, Supplementary Fig. 13). In general, all tools were consistent with the overall WGBS pattern independently of coverage and CG content (Fig. 5). All tools recapitulated the known pattern of hypo-methylation at CpG islands (CGIs)[20] (Supplementary Fig. 13). However, there were local inconsistencies with some of the tools. For instance, in the region spanning the first and second introns of *IRF4* (chr6:392228-401463) (Fig. 5a), Guppy and DeepMod underpredicted the methylation frequency. In the region spanning the genes *ACKR1* and *CADM3* (chr1:159199780-159212236), Tombo overpredicted the methylation frequency, and Guppy failed to predict an increase in methylation described by WGBS and the other tools (Fig. 5b). At the *TPO* locus (Fig. 5c), the intermediate methylation at a CGI described by WGBS was recovered by all tools, except Tombo, Nanopolish, and METEORE (RF) where they overpredicted. Using the eight different regions from Gilpatrick et al.[18], we found similar results (Supplementary Fig. 14). For instance, Guppy and DeepMod underpredicted relative to WGBS in the region of *GPX1* promoter (chr3: 49352525-49366169), whereas Tombo overpredicted at a CGI in the same region (Fig. 5d).

Next, we applied those thresholds derived before from the individual read analysis for different methods on the two independent nCATS data. Using a single score cutoff based on the maximum value of TPR–FPR (Supplementary Table 3), we observed slight improvements in correlation and RMSE values for DeepSignal, Tombo, and METEORE (RF) in both nCATS datasets (Supplementary Tables 9 and 10). Using the strategy of a double cutoff based on the removal of the 10% of reads that were closest to the score at which the FPR and 1-TPR curves crossed (Supplementary Table 4), the concordance with WGBS improved for DeepSignal, Tombo, and METEORE (RF) in both datasets, and for Nanopolish only in our experimental dataset (Supplementary Tables 11 and 12). The results showed signs of

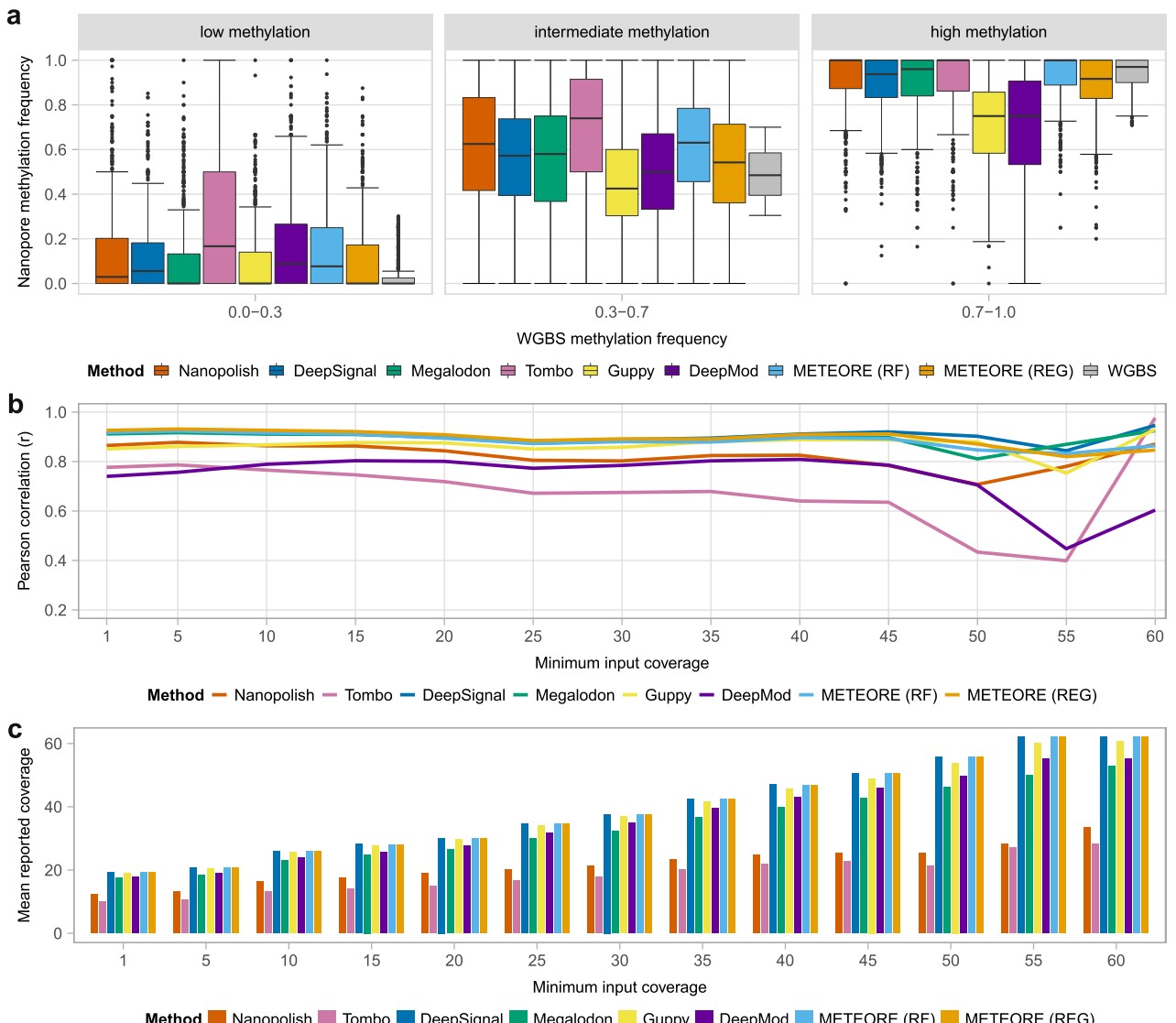

**Fig. 4 Comparison of CpG methylation predictions from nanopore with whole-genome bisulfite sequencing (WGBS). a** Distribution of methylation calls ($n = 1743$) from nanopore ($y$ axis) across three WGBS methylation bins: no or low methylation (0.0-0.3) ($n = 793$), intermediate methylation (0.3-0.7) ($n = 264$), and high or full methylation (0.7-1.0) ($n = 686$). In the boxplots, the lower and upper boundaries of the box are the first and third quartiles of the data, respectively, with the median indicated by a thick black line. The lower and upper whiskers extend to 1.5 times the interquartile range. The outliers are represented by the black dots. **b** Pearson's correlation ($r$) ($y$ axis) between methylation frequencies calculated from nanopore reads and WGBS at sites by each of the tested tools (combining predictions from both strands) at each level of minimal input coverage, i.e., minimum number of nanopore reads considered per site as reported from the BAM file ($x$ axis). **c** Mean reported coverage (using the coverage reported by each tool for all sites) at each value of minimum input coverage in **b**. METEORE (RF) is the combination of Megalodon and DeepSignal using a random forest (parameters: max_depth = 3 and n_estimator = 10), and METEORE (REG) is the combination of Megalodon and DeepSignal using a regression model.

improvement in some methods such as DeepSignal, Tombo, and METEORE (RF) by using an alternative cutoff, whereas Nanopolish, Megalodon, Guppy, and METEORE (REG) showed no improvement in accuracy. This might be due to the scores provided not being suitable for the prediction task (Supplementary Fig. 4), or to the high variability of the nanopore signal across different sequence contexts. This prompted us to study the potential sequence biases associated with correct and incorrect predictions.

**Sequence context biases in the capability of methylation prediction.** To explore the effect of the sequence context in the capability of methylation prediction, we calculated the absolute differences between methylation calls from nanopore reads by each tool and the methylation calculated from WGBS for each CpG site in an 8-mer context (NNNCGNNN). We observed that despite the variability between methods, there is a subset of k-mers where all methods have low or no discrepancy with WGBS, whereas for other k-mers all methods show a high discrepancy (Supplementary Fig. 16a). Looking at the top and bottom 40 8-mers according to the average discrepancy across methods, we observed that nanopore and WGBS disagreed the most in an AT-rich context (Supplementary Fig. 16b). To further investigate these biases, we grouped the k-mers according to the GC content. Nanopolish, DeepSignal, Megalodon, and METEORE showed similar agreement with WGBS across all GC content, whereas Guppy and DeepMod underperformed compared to other tools,

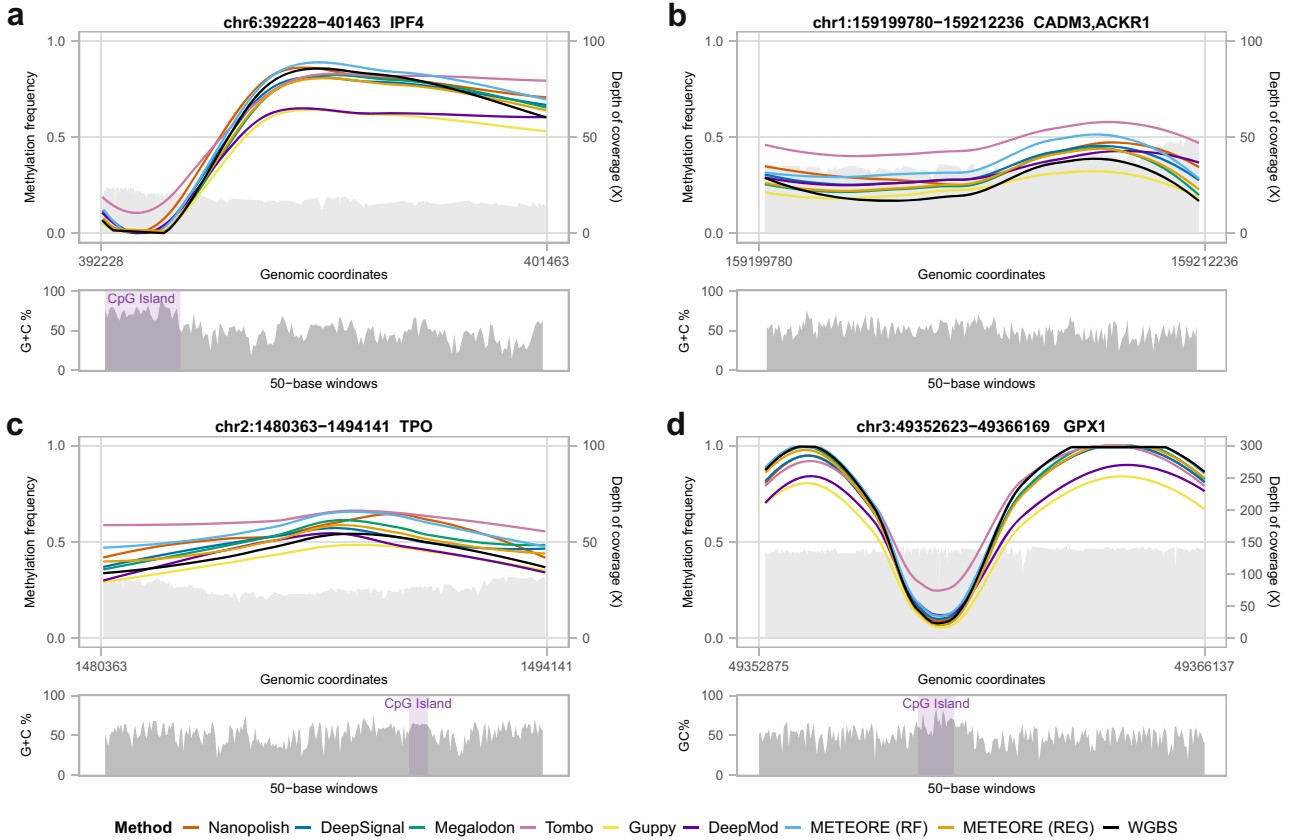

**Fig. 5 Comparison of CpG methylation predictions from nanopore with whole-genome bisulfite sequencing (WGBS) along Cas9-targeted regions.**
Locally estimated scatterplot smoothing (LOESS) line plots of methylation calls frequency predictions (left *y* axes) from WGBS Illumina and from nanopore data using seven tools: Nanopolish, DeepSignal, Megalodon, Tombo, Guppy, DeepMod, and METEORE random forest (RF) and regression (REG) models. The plots include the nanopore read coverage (right *y* axes), shown as a light gray area. The panels below show the GC content of the region, using a window size of 50 bases. **a–c** Three of our ten target regions and **d** one of the eight target regions from Gilpatrick et al.[18]. The depicted regions are **a** chr6:392228-401463, which covers the first and second introns of gene *IRF4*; **b** chr1:159199780-159212236, which covers the genes *ACKR1* and *CADM3;* **c** chr2:1480363-1494141, which covers the gene *TPO*; and **d** chr3:49352525-49366169, which covers the genes *GPX1*. A zoom in of the LOESS line plots with the individual methylation calls and logos for the sequence context around the CpG site in four CpG islands (CGIs) of our target regions are shown in Supplementary Fig. 15.

especially at low GC content, and Tombo showed higher dispersion at intermediate GC content (Supplementary Fig. 16c). To investigate the specific biases for each tool, we grouped the k-mers according to the low (=0) or high (>0.5) absolute difference with WGBS. By assessing the differences in the sequences of "high" and "low" k-mers for each tool, we observed that Nanopolish, Megalodon, and METEORE showed no sequence biases (Supplementary Fig. 17). In contrast, DeepSignal showed a bias for high discrepancy for a T at the 1st position and an A at the 8th position of the 8-mer, Tombo and Guppy were affected by the residues at the 1st (A for Tombo and T for Guppy) and 6th (T for Tombo and G for Guppy) positions, whereas DeepMod had many k-mers with a high discrepancy that with a significant bias for A and T (Supplementary Fig. 17).

## Discussion
Our systematic benchmarking of DNA methylation prediction from nanopore sequencing using individual reads, controlled methylation mixtures, Cas9-targeted sequencing, and WGBS, indicated that no single method predicts correctly across all ranges of methylation frequency. Extreme cases were Guppy, which correctly identified unmethylated sites but failed at fully methylated sites, as well as Nanopolish and Tombo, which were

able to recover fully methylated sites but had a high rate of false positives at unmethylated sites. Moreover, the predictions per site generally showed a high dispersion and did not generally agree with the expected methylation frequency. A possible reason for this low agreement is that the M.SssI treatment used to generate the CpG-methylated control only has 95% efficiency[8]. Although we used 10% incremental windows to accommodate for this potential variability in our tests, the dispersion occurred across all methods and methylation levels. In addition, when we focused on the most reliable sites for either 0 or 100% methylation[8], the tested tools still presented dispersion as observed before (Supplementary Fig. 18). This indicates that there is still a major limitation in the capacity to correctly recover the methylation levels from individual reads.

The observed low agreements motivated us to propose a consensus approach, METEORE, aiming to ameliorate the deficiencies from some methods and incorporate the advantages from others. The combination of the two methods improved the overall accuracy at the levels of individual reads and per-site methylation frequency. Our analyses thus suggested that it is generally advantageous to run at least two or more tools to obtain an accurate picture of the DNA methylation patterns. Although combining multiple tools improved the accuracy even further, it might be impractical for routine analyses, due to the running

times for some methods without GPU support (Supplementary Table 12). Megalodon provided overall the best performance but relies on GPUs to be time efficient. In contrast, Nanopolish and Tombo were the fastest on CPUs but showed the largest dispersions and FPRs. The combination of Megalodon and DeepSignal struck a good balance for accuracy and running times but would require a GPU for efficiency. On the other hand, on a CPU the combination of Nanopolish and DeepSignal can achieve an accuracy like Megalodon and be time competitive (Supplementary Table 13). In addition, we found that by reassessing the score cutoffs for individual reads, the per-site methylation predictions could be improved in some methods. Furthermore, there was an advantage in removing sites with uncertain methylation status. This strategy improved the accuracy of some methods. However, it had a large impact on the number of reads for Nanopolish and Tombo, which discarded more reads than the rest of the methods.

The comparison with WGBS datasets using two independent nCATS experiments recovered results consistent with the analyses with individual reads and control mixture datasets. Although the tools recovered the overall WGBS patterns across different genomic regions, and independently of GC content and coverage, there were some remarkable variations. In particular, Nanopolish tended to overpredict methylation values, DeepMod and Guppy tended to underpredict, and Tombo and Guppy showed local discrepancies with the other tools. Megalodon, DeepSignal, and the consensus of these two methods using METEORE achieved overall good consistency with the WGBS data. Furthermore, we observed a limited impact of coverage on accuracy. This suggests a strong potential for the development of sensitive diagnostic and forensic tests without relying on high coverage. Moreover, the use of our improved strategies with single or double cutoffs in some methods such as DeepSignal and Tombo or with METEORE consensus predictions will further facilitate accurate analyses using individual reads. Lastly, we observed that the highest discrepancy with WGBS occurred at CG sites in the context of AT-rich sequences (Supplementary Fig. 16b). This bias was particularly strong for DeepMod and DeepSignal, whereas there were no significant sequence biases for Nanopolish, Megalodon, and METEORE related to the discrepancy with WGBS. Although these discrepancies could be due to the detection differences between nanopore and WGBS, they could also point to the disparate types of models and training regimes used by the tools tested.

In summary, we highlighted the strengths and weakness of state-of-the-art methods to predict DNA methylation from nanopore sequencing and provided various strategies to improve the prediction accuracy. We expect METEORE and the provided pipelines will facilitate the accurate analysis of genome-wide methylation patterns both per site and in individual molecules in multiple biological contexts.

## Methods

**DNA methylation of controlled mixtures**. We used *E. coli* (K12 ER2925) control reads[8], for which DNA was amplified by PCR (unmethylated, negative control) and half of the PCR-amplified DNA was subjected to CpG methyltransferase (M.SssI) treatment to methylate cytosines at CpG sites (methylated, positive control). It was reported that M.SssI has an efficiency of 95%[8]. This would not affect the benchmarking with the fully unmethylated set (0% methylated) but it might affect the other mixture datasets. Accordingly, to avoid potential biases in our benchmarking analysis, we considered bins of percentage methylation ranges, in steps of 10%. For instance, the bin of the fully methylated sites included sites of 90–100% methylation. After alignment of all PASS reads for both controls using Minimap2[21] and removing secondary and supplementary reads, 110,795 reads of unmethylated control and 69,453 reads of methylated control remained. From the 346,793 CG sites in the *E. coli* reference genome (NC_000913.3), we selected 100 random sites with a single CpG in a 20 nt window (NNNNNNNNNCGNNNNNNNNNN, with no CG in the region with Ns) that had aligned sequencing reads from both positive and negative control datasets. Using the filtered PASS reads covering these 100 selected CpG sites from the control datasets (Supplementary Data 1), we

created 11 benchmarking datasets with specific mixtures of methylated and unmethylated reads, namely, containing 0%, 10%,…, 90%, 100% of methylated reads, which we used for the initial benchmarking of the tools to predict different methylation mixtures per genomic site (mixture dataset 1) (Supplementary Data 3). For independent validation, we built an independent mixture dataset 2. We selected a different set of 50 CG sites (single CG in a 20 nt window) (Supplementary Data 2), and further selected 6803 different reads from the remaining fully methylated or fully methylated reads not used in the dataset above. We built again 11 benchmarking datasets with specific mixtures of methylated and unmethylated reads, namely, containing 0%, 10%, 20%, …, 90%, 100% of methylated reads (mixture dataset 2) (Supplementary Data 4). We used the mixture dataset 2 to obtain the correlation of the different methods with the expected percentage methylations using the default thresholds suggested by each tool and alternative thresholds obtained from dataset 1. We also used this dataset to test our combined model (see below).

**CpG methylation detection with six different tools**. We developed a standardized workflow for six tools: Nanopolish[8], Megalodon[12], DeepSignal[13], Guppy[14], Tombo[15], and DeepMod[16]. Snakemake pipelines to run these tools are available at GitHub[11]. When we selected the tools to be included in this study, we excluded NanoMod[22], as it only allowed detection of methylation differences using a control and a test sample, and SignalAlign[10], as its repository had not been updated for over four years; and mCaller[23], because it only had been trained for 6mA, but not for 5mC.

Methylation level was collected for individual CpG sites in both strands and considered per site and per read or summarized (methylation frequency) per site. For the comparison with bisulfite sequencing, we also considered the approach of merging the CpG methylation calls from both strands into a single strand. This was done in the following way: For the sites that had methylation frequencies from both strands, we combined the sites by averaging the methylation frequencies and adding up the coverage. Otherwise, we kept the methylation frequency the same if the site only had methylation prediction from one strand.

Nanopolish (v0.13.2) assigns a log-likelihood ratio to each individual CpG site or to a group of nearby CpG sites that share the same methylation level in each site within the group. Nanopolish uses reads with a minimum mapping quality score of 20. A positive log-likelihood ratio value indicates evidence of methylation. To include all the predictions per read and per site, such CpG groups were split up into the constituent sites with the same log-likelihood ratio using a Python script incorporated in our Snakemake pipeline (https://github.com/comprna/METEORE)[11]. The output file was further processed in R (v3.6.3). As for default cutoffs, we considered those originally suggest by Simpson et al.[8], i.e., log-likelihood > 2.5 for methylated sites and <−2.5 for unmethylated sites. Although the default log-likelihood ratio threshold has changed to 2.0 in v0.12, we did not see major differences between both thresholds (Supplementary Fig. 19). For our benchmarking analysis we used the log-likelihood ratio of 2.5 for sites in individual reads. The methylation frequency was then calculated for each site as the number of mapped reads predicted as methylated divided by the number of total mapped reads.

Tombo (v1.5.1) resquiggles all raw nanopore reads before modified base detection. It then implements three approaches to detect nucleotide modifications: (1) the alternative base detection approach, which computes a statistic by scaling log-likelihood ratios to identify targeted bases where the signals match the expected level for a non-canonical base; (2) the de novo approach, which performs a hypothesis test by statistically comparing signals to an in-silico reference; and (3) the sample comparison approach. This latter approach provides two different ways for modified base detection, one uses a canonical model adjusted by a control set of reads to identify deviations between expected and observed levels, while the other one compares signal levels from two sets of reads at each reference position. Of the three approaches, we used the alternative model specific for CpG to be able to predict CpG methylation in individual samples and reads. This model tests the signal levels against expected canonical and alternate 5mC in at CG motifs, producing the per-read binary statistics in HDF5 format, where positive values indicate canonical bases and negative values modified bases. For the initial analysis, we used the default cutoffs of −1.5 and 2.5 where scores below −1.5 were considered as methylated and above 2.5 unmethylated, and scores between these thresholds did not contribute to the per-site methylation. Tombo outputs four individual wiggle files (WIG format), one per strand, reporting the read coverage level and the methylation scores. These files were converted to the same TSV format used by other tools with a Python script included in our Snakemake pipeline, with the score and coverage for each mapped CpG site for downstream analyses. For the benchmarking analysis per site and per read, we used the per-read binary statistics given by Tombo.

DeepSignal (v0.1.7) uses the resquiggling algorithm from Tombo in the first step. It then predicts the methylation state of the targeted cytosine at CpG motifs and outputs the probabilities of being methylated, P(m), and unmethylated, P(u), for each cytosine in each read. For the initial analysis, a base was considered as methylated if the methylated probability was greater than the unmethylated probability, P(m) > P(u), and unmethylated otherwise. In addition, a Python script was added to our Snakemake pipeline to calculate the methylation frequency at

each CpG site. For the benchmarking analyses per site and per read, we used a score calculated as the log2 ratio of the probabilities, i.e., $\log_2(P(m)/P(u))$.

Guppy (v3.6.0) filters the nanopore reads based on the read alignment and implements a neural network model that basecalls 5mC at CG sites as a fifth base along with the four canonical DNA bases. It is only available to members of the nanopore community (https://community.nanoporetech.com). We used the configuration file named dna_r9.4.1_450bps_modbases_dam-dcm-cpg_hac.cfg to run running Guppy's modified basecalling model. This produced the reads supporting the modifications in fast5 format. We then used the scripts provided at https://github.com/kpalin/gcf52ref to convert the guppy methylation calls from fast5 files to reference anchored files similar to Nanopolish. These scripts were incorporated into our Snakemake pipeline (https://github.com/comprna/METEORE)[11].

Megalodon (v2.2.4) implements a neural network model that uses Guppy (v4.0.11) to rebasecall all reads and then identifies 5mC by anchoring the basecalling output to the reference, assigning a score for the candidate modified base and performing a calibration for the conversion of the raw scores to estimated empirical probabilities. Megalodon requires Guppy to be installed and the path to Guppy basecalling executable server to be set. We used the most recent basecalling model in Rerio for Megalodon[24] (res_dna_r941_min_modbases_5mC_CpG_v001.cfg). Megalodon produces per-read modified base log probability and canonical base log probability at each mapped CpG site. A default threshold of 0.75 was used as a minimum score for both modified and canonical base probabilities to include a modified basecall in the final aggregated output in the bedMethyl format file containing per-site coverage and methylation percentage. For the benchmarking analysis per site and per read, we used a log-likelihood score calculated by subtracting the natural log probability of the modified base, log(M), and the natural log probability of the canonical base, log(C), resulting in log(M/C).

DeepMod takes single-read fast5 files as input and uses reads with a mapping quality score greater than 10. It then outputs a methylation prediction summary per site at genome level for each strand in a BED format containing the coverage, the number of methylated reads, and methylation percentage. As DeepMod did not provide any information for individual reads, we could not use it for the per-read benchmarking analyses.

**METEORE consensus models**. METEORE was created to provide a consensus prediction using the scores from two or more methods. As the two main approaches for supervised learning are classification and regression, we implemented both types of methods in METEORE to test the combination of tools. For the classification model, we used an RF classifier[25] from the Python sklearn library[26]. We scaled the prediction scores from each individual method to the range of [0, 1] using min–max scaling[25]. The ROC and PR curves were built from a tenfold cross-validation on the prediction scores from the input methods to produce ROC and PR curves. The reads were randomly selected during cross-fold validation. For the METEORE implementation, we used the parameters max_dep = 3 and n_estimator = 10. We also tested the default parameters of RF from sklearn (n_estimator = 100 and max_dep = None). The METEORE RF model tested in our analyses was trained on the entire mixture dataset 1 with the scores from Megalodon and DeepSignal and using parameters max_dep = 3 and n_estimator = 10. The multiple linear regression-based approach used sklearn's RidgeCV linear regression (REG) model, and min–max scaling as in the RF model. The initial ROC and PR curves were produced with the built-in fivefold cross-validation in RidgeCV. The METEORE REG model tested in our analyses was trained on the entire mixture dataset 1 with the scores from Megalodon and DeepSignal. After prediction, resulting scores were classified as unmethylated if they were less than 0.5 and methylated if greater. METEORE REG required reads with prediction scores from all provided tool inputs for using multiple linear regression to model expected methylation, leading to a slight (~10%) decrease in observed reads in each test. The scripts to train and run METEORE are available at https://github.com/comprna/METEORE[11].

**Processing of per-site methylation calls**. The raw output of each of the tools contained methylation information for each aggregated CpG site on both strands. That is, each mapped CpG site on the positive strand of the human reference genome had a counterpart CpG site mapped on the negative strand. To perform the per-site benchmarking, we used the positive strand coordinate system, so the mapped sites that were on the negative strand were lifted to positive coordinates by subtracting 1 from the coordinate position. If there were predictions made on both strands for the same site, we obtained the mean methylation frequency for that site. If there was no per-site information only for one of the strands, we kept the same prediction from that strand and lifted it to the positive strand if necessary.

**Sequence motif analysis and visualization**. We collected 7-mers of targeted CpG sites from *E. coli* mixture datasets and all sites detected in the CGIs from the nCATS datasets. Sequence logos were generated from 7-mer data in FASTA format using WebLogo[27]. We explored the k-mer contexts of the CpG sites using the high-coverage nCATS data from Gilpatrick et al.[18]. The sequence motifs were grouped by the absolute difference between the methylation frequencies produced by each tool and the WGBS values for each CpG site in an 8-mer context (NNNCGNNN),

where a site with a score of 0 was labeled as "low discrepancy," and with a score greater than 0.5 was labeled as "high discrepancy." The pLogo web tool[28] was used to visualize the sequence motifs and assess the significance of the differences in the frequency of residues between the "low discrepancy" k-mers (foreground dataset) and the "high discrepancy" k-mers (background dataset).

**Guide RNA design and RNP complex assembly**. We used the nCATS protocol[18] to target ten regions of the human genome. This PCR-free protocol uses Cas9 to cut double-stranded DNAs (dsDNAs) at specific sites and then preferentially ligates sequencing adapters to the cleaved ends for enrichment[18]. The cleaved target DNA strands with adapters attached are then sequenced. We designed ten pairs of RNA CRISPR guides (crRNAs) for ten forensically relevant regions (Supplementary Table 5). An initial panel of candidate crRNAs was designed using the freely available tool CHOPCHOP[29] as recommended in the ONT protocol. For each target region, about three to five best crRNAs were selected based on the cleavage location, crRNA efficiency, and the number of predicted mismatches using CHOPCHOP. The crRNAs were then evaluated by the IDT's design checker[30] to select for high on-target performance and low off-target activity. Ten pairs of guide RNA were used to enrich ten human regions ranging from 8 to 36 kb. Here, one gRNA was used on either side of each target region (Supplementary Table 6). Each RNA oligo including crRNAs (IDT, custom-designed) and tracrRNA (IDT, 11-05-01-12) was resuspended in IDTE buffer pH 7.5 (IDT, 11-01-02-02) to a final concentration of 100 μM. crRNAs were then pooled to make an equimolar crRNA mix by combining equal volumes of each crRNA. In order to form gRNA duplexes, the crRNA pool and tracrRNA were then combined in equimolar concentrations with duplex buffer (IDT, 11-05-01-12), followed by denaturation for 5 min at 95 °C, then allowing to cool to room temperature. RNP complexes were created by assembling the following components: gRNA duplexes, CutSmart buffer (NEB, B7204), nuclease-free water, and Cas9 endonuclease (IDT, 1081058).

**Library preparation**. Genomic DNA (gDNA) from the GM12878 human cell line was obtained from the Coriell Institute (coriell.org) (cat. no. NA12878). The purity of the purchased gDNA was measured with the Nanodrop spectrophotometer (Thermo Fisher) at the 260/280 and 260/230 nm values. Dephosphorylation of 5′ ends of DNA was performed as follows. A 5 μg amount of input DNA was dephosphorylated to prevent downstream adapter ligation using Quick Dephosphorylation Kit (NEB, M0508). DNAs were resuspended in the CutSmart buffer and dephosphorylated with Quick CIP enzyme in a PCR tube for 10 min at 37 °C, followed by heating for 2 min at 80 °C for CIP enzyme inactivation. Cas9 Cleavage and dA-tailing was performed as follows. a 100 mM aliquot of dATP (NEB, N0440S) was first diluted to 10 mM. After allowing the dephosphorylated DNA sample to return to room temperature, the preassembled RNP complexes, 10 mM dATP, and Taq DNA Polymerase with Standard Taq Buffer (NEB, M0273) were added to the PCR tube containing the sample for the in vitro digestion reaction and subsequent dA-tailing. The sample was then incubated at 37 °C for 15 min, and then at 72 °C for 5 min. By dephosphorylating preexisting DNA ends prior to Cas9 cleavage, sequencing adapters and ligation buffer from the Oxford Nanopore Ligation Sequencing Kit (ONT, LSK109) were preferentially ligated to the cleaved DNA ends at Cas9 cleavage sites using T4 Ligase from the NEBNext Quick Ligation Module (NEB, E6056) for 10 min at room temperature. The sample was cleaned up to remove excess adapters using the Agencourt AMPure XP beads (Beckman Coulter, A63881), washing twice on a magnetic rack with the long-fragment buffer (ONT, LSK109) before eluting in 14 μl of elution buffer (ONT, LSK109). A 1 μl aliquot of the final library was quantified using the Qubit dsDNA Broad Range Assay Kit (Thermo Fisher). Starting with 5 μg of input DNA, 1 μg of DNA was recovered after library preparation. The library was stored on ice until ready to load.

**Sequencing and data processing**. Before loading, the flowcell was primed with a solution consisting of flush buffer (ONT, LSK109) and flush tether (ONT, LSK109). The sequencing library was prepared by adding sequencing buffer (ONT, LSK109) and loading beads (ONT, LSK109) into the DNA library, and then loaded into the flowcell. The sample was run on a MinION flowcell (FLO-MIN106, R9.4.1 pore) using the MinION sequencer for 19 h, operated using the MinKNOW software. Live basecalling was carried out during the experiment using Guppy's fast basecalling model in MinKNOW. The resulted FASTQ files were immediately aligned to the human reference genome (GRCh38/hg38) using minimap2[21], followed by visualization with Integrative Genomics Viewer[31] to confirm the generation of on-target sequencing reads. Post-run basecalling was performed using Guppy (v.3.2.4) high-accuracy model to generate the final set of sequencing reads with higher read accuracy than the fast model and recognition of modified bases from the electrical signal data. Reads were aligned to the human reference genome (GRCh38/hg38) using minimap2. Using Samtools[32] (v1.9), we collected aligned reads within the enriched regions as on-target reads. Those outside the targeted regions were considered off-target reads and subsequently discarded. Coverage plots for target regions were generated using the Snakemake[17] workflow developed by Oxford Nanopore Technologies[33].

**Comparison with bisulfite sequencing data**. We compared with the published WGBS data[19] for NA12878 (ENCODE accessions: ENCFF279HCL, ENCFF835NTC) using two different approaches. In the first approach, we compared every CpG site on both strands for nanopore and WGBS data, preserving the strand information. In the second approach, we used the positive strand coordinate system by lifting all sites from the negative strand to be on the positive strand, i.e., site position on the negative strand minus one. For the sites that had methylation evidence from both strands, we combined the sites by averaging the methylation frequencies and adding up the coverage. We preserved the information if the site only had methylation prediction from the positive strand or the negative strand.

To obtain high confidence methylation calls from WGBS data for validation, the resulting individual or combined CpG sites were processed in the following way. CpG sites with zero coverage from both WGBS replicates were discarded. Furthermore, we calculated the difference in methylation frequency between both WGBS replicates and considered the 0.1 and 0.9 quantiles of the distribution of differences. A CpG site was kept if the difference for that site was between those 0.1 and 0.9 quantiles, otherwise it was removed. We finally calculated the number of sites, Pearson and Spearman correlations, and coefficient of determination between methylation frequencies calculated from WGBS and those calculated from nanopore reads by each of the tested methods using cor.test() and lm() functions from R[34]. We also subsampled the reads for the CpG sites that were covered with at least 1, 5, 10, 15, 20, etc. reads and calculated Pearson correlation at different levels of read coverage for further evaluation.

**Reporting summary**. Further information on research design is available in the Nature Research Reporting Summary linked to this article.

## Data availability

Nanopore sequencing data generated in this study have been deposited in the Sequence Read Archive (SRA) under study accession PRJNA656260. Nanopore sequencing data from Gilpatrick et al.[18] used in this study are available in SRA under study accession PRJNA531320. Nanopore sequencing data for *E. coli* methylated and unmethylated genomes used in this study are available at the European Nucleotide Archive study accession ERP014559[8]. WGBS data from the ENCODE project[19] used in this study are available at https://www.encodeproject.org/ under IDs ENCFF279HCL and ENCFF835NTC. CpG sites used in both control mixture datasets are provided in Supplementary Data 1 (mixture dataset 1) and 2 (mixture dataset 2). Raw nanopore sequencing data in fast5 format of *E. coli* methylated and unmethylated control samples[8] that were used to generate both mixture datasets are provided in Supplementary Data 3 (mixture dataset 1) and 4 (mixture dataset 2). Predicted methylation frequencies of CpG sites for each tested method in mixture datasets 1 and 2 are provided in Supplementary Data 5 and 10, respectively. The number of true negatives and true positives for each tested method in accordance with different thresholds used to call a site unmethylated/methylated are provided in Supplementary Data 6 and 7, respectively. Performance comparisons of five tested tools in read level are provided in Supplementary Data 8 (ROC curves) and 9 (PR curves). Methylation frequencies and coverage of all CpG sites reported by each tested tool across the target regions for the nCATS datasets are provided in Supplementary Data 11 (our data) and 12 (Gilpatrick et al.[18]). For k-mer contexts analysis, data were provided in Supplementary Data 13.

## Code availability

METEORE is available at https://github.com/comprna/METEORE[11] under the MIT license.

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

## Acknowledgements

The authors would like to thank Jared Simpson for making the *E. coli* K12 ER2625 data publicly available, Timothy Gilpatrick for making the Cas9-targeted nanopore data of NA12878 publicly available, Marcus Stoiber and Marta Verdugo from Oxford Nanopore Technologies for constructive comments, and Niccy Aitken from the EcoGenomic Facility, Research School of Biology (Australian National University) for providing help and support on library preparation and sequencing.

## Author contributions

E.E. conceived the study. E.E. and C.J. supervised the whole project. Z.W.-S.Y. and E.E. designed METEORE. Z.W.-S.Y. developed pipelines of METEORE. A.S. built the RF model for the consensus predictions of METEORE. C.J. built the regression model for the consensus predictions of METEORE. R.D. and D.M. advised on the experimental design. Z.W.-S.Y. performed the sequencing experiments, methylation detection, and benchmarking analyses. E.E., C.J., Z.W.-S.Y., and A.S. contributed to the interpretation of the results. Z.W.-S.Y. and E.E took the lead in writing the manuscript with input from C.J., A.S., R.D., and D.M.

## Competing interests

The authors declare no competing interests.
