## [Peer Review File · Nature Communications]

Reviewers' Comments:

Reviewer #1:

Remarks to the Author:

In this study, the authors present a systematic benchmarking of six tools for the detection of CpG methylation using Nanopore sequencing. Based on the evaluation, they also proposed a consensus approach (METEORE) to improve accuracy over individual tools. The benchmarking includes evaluation using individual reads, control mixtures of methylated and unmethylated reads, and Cas9-targeted sequencing and whole-genome bisulfite sequencing. The evaluation of the tools can help understand and improve the prediction of CpG methylation using Nanopore reads. I have two major concerns: Major Concerns::

1. The performance of METEORE, which combined DeepSignal and Megaloddon, did not improve much comparing with those of original DeepSignal and Megaloddon tools.
2. The evaluation of the correlation of CpG prediction using WGBS data and different coverages of Nanopore reads did not make sense. What the authors used is of the different levels of coverages, but the coverage > minimum coverage. Figure 4, supplemental Figure 9 and 10 using "Nanopore coverage" as X-axis is confusing. With more coverage of Nanopore reads, the correlation of CpG prediction using WGBS and Nanopore reads should increase. The performance decreasing in Figure 4, supplemental Figure 9, and 10 are due to the fewer CpG reads covered by high coverage reads. To demonstrate the correlation of CpG prediction using WGSB and different coverage of Nanopore reads, the authors should sample different coverage of Nanopore reads and make the prediction, as did in DeepSignal paper.

some minor comments:

1. On page 4 "p-value < 2.2.e-16 for all tools", how were the p-values calculated?
2. On page 4, "Supp. Table 2" was quoted 2 times, and it seems that either the two quotes are the same (there is no need to quote/repeat the sentence twice), or one of them is not quoted appropriately.
3. On page 17, "bi-sulfite" should be "bisulfite", to be consistent with the whole manuscript.
4. On page 18, the methylation LLR threshold of Nanopolish has changed from 2.5 to 2.0 since v0.12.0. The authors should check the actual threshold used in their analysis.
5. The analysis results of Megaloddon in Supp. Fig. 8b and Supp. Table 7 are not the same. The authors should check which one is right.
6. How did the N in Table 1 obtain?

Reviewer #2:

Remarks to the Author:

I have no comments on the paper. Its a nice study on comparison of tools for detection of methylation using nanopore sequencing data. The findings are reasonable and I believe would be useful to sequencing centers and researchers in this area.

Reviewer #3:

Remarks to the Author:

In this manuscript, the authors set out to benchmark software for calling 5-methylcytosine (5mC) base modifications from Nanopore data. They compare six state-of-the-art software tools, and, unsatisfied with the accuracy provided, created their own 5mC calling pipeline (METEORE) by combining two or more of these six tools. Then then went on to evaluate this against the other tools in both the E. coli genome and targeted locations in the human genome.

Notably, the authors:

- A. Created a pipeline that standardizes inputs and outputs so that different tools can be easily compared. This also helps METEORE to be more effectively used in the future.
- B. Created a clever test set for their bakeoff analyses but using publicly available "100% methylated" reads to create different methylated proportions.
- C. Performed comparisons between the tools including how different methylation calling cutoffs affect tool performance.
- D. Applying their tool and their analyses to human data.

However, I still have some significant problems with the analysis:

1. Small point - megalodon is also a neural net based model - after anchoring the data to the reference, the tool uses the neural net Guppy output and viterbi decoding to find the best path to classify modifications (if I'm not mistaken)
2. Figure 2 was . . . hard to parse, I think because of the number of different things being plotted on the same graph. 2A for example it is easy to see the dispersion (of nanopolish for example), but hard to see the linearity. I'd consider breaking this into separate plots for clarity, or putting some of it in the supplement to make the most critical points. A box-plot instead of violin may allow more clarity. 2b-d I'd suggest some sort of ecdf plot or other line based plot instead of bars - the bars just take up space when you could have lines to show the data just as well.
3. Are the authors using map quality scores for filtering the aligned reads - this is a stringent cutoff in some tools, but not in others and could easily affect the quality of the resulting calls.
4. Choice of CpGs for analysis:
 - a. The majority of the benchmarking in the manuscript is based on 100 "random" CpGs from the E. coli genome. As the authors are no doubt aware, nanopore sequencing derives its signal from a k-mer based model, with (presumably) some k-mers performing better than others. The size of the signal window is even different between the different tools. This can result in some tools performing better and worse in different areas. This is exemplified by the fact that the authors found different numbers on the second set of 50 CpGs, but the k-mer contexts of these CpGs are not clear (comparing Figure 2A to Figure 3E).
 - b. While some of these differences are small, the authors use small differences to claim that their tool outperforms all other tools ("METEORE RF combining Megalodon and DeepSignal achieved lower RMSE compared with the individual tools (Fig. 3e)"; page 9). For example the difference between DeepSignal for mixture 1 and mixture 2 is larger than the difference between METEORE RF and Megalodon in mixture 2. It is not clear to me why the authors can't analyze the entire singleton set of the E. Coli genome?
 - c. The CGs discussed are also profiled by Illumina bisulfite sequencing data in Simpson et al, with that data provided in ENA, the authors should consider examining the exact bisulfite provided methylation levels rather than estimating since they are looking at specific CGs.
 - d. Failing this, or even in preference to it, Simpson et al also contains a PCR amplified and M.SssI treated GM12878 control set which could provide another control set with even more k-mer contexts. This might be useful to demonstrate the performance on a human genome - the putative end goal. I know the authors used the targeted human data later to evaluate, but this is not the same as a more agnostic evaluation (more on the targeted data later)
 - e. The authors specifically test the performance of their tool on singleton CGs, i.e. CGs with no other CG within 10 bases up or downstream. This is fair to eliminate signal from other nearby CGs, but presumably is a very different evaluation than operation on a human genome, with relatively dense CGs in CGI (for example).
 - f. This all fits into the point: Identifying k-mers where a tool performs better or worse would be very useful.
5. The authors have developed a combination of tools to call methylation - trying to improve on the weaknesses of any given tool with either linear regression or random forest on the outputs. Ideas like this already exist for structural variants (e.g. Parliament), but in theory this could provide improved accuracy in methylation calling. But the authors do not assess concerns of runtime/resources; many of the tools the authors list already have substantial runtime/hardware requirements - running more than one provides an additional burden. And seemingly for a small incremental improvement over

megalodon alone. The authors need to make a clear point that adding another tool (even just deep signal) is "worth" the effort, because that isn't clear from the current results. This should _especially_ be benchmarked on larger genomes - though the targeted sequencing is certainly a valuable application, whole genome methylation will also likely be an important application.

6. I am confused by the optimum cutoff settings, especially in terms of Megalodon (the tool with the closest performance to METEORE). While the cutoffs improved general metrics of methylation calling, when you compare methylation percentages outside of the windows in Fig. 3f to Supplemental Fig. 3c/3d, the optimized cutoffs for many of these tools made the tools _worse_ than default settings. Can you explain how an "optimized" setting would make the calls worse? It would be a worthwhile point to explain this in the paper as well.

7. It is not 100% clear, but it seems in the bisulfite work that the authors are no longer requiring that the CGs be distant from each other? Given that it is more relevant to plot local CG density rather than just GC% per se.

8. I like the comparison of METEORE to WGBS. However, I again believe that the authors should expand the analysis on the data at hand and dig into _where_ (i.e. what k-mers) have differences beyond just correlation and broad patterns at promoters. This would likely be very informative for those in the community going forward to understand the limitations of these tools. In this vein, it would be worthwhile to better visualize WGBS methylation percentages in Figures 5 and Supplemental 11/12. It is difficult to see (grey lines) and that is probably the most relevant comparison in those figures.

9. I'd be more interested in a faceted scatter plot of methylation versus bisulfite than the overall "binned" methylation shown in Figure 4a - I think the bins lose subtlety.

10. On 4b: It seems like using the "filtered" read coverage for nanopolish and tomo provides a bit of a skewed picture of the coverage required to achieve correlation, since in reality you need a higher coverage (~60X) to get the 35X nanopolish coverage reported? I would suggest reporting the coverage of data _input_ into nanopolish instead.

11. More on 4b - it seems that there is a big dip in correlation of both METEORE modes at 55X - and generally that higher coverage gives worse results? The authors do not offer an explanation for this, but I'd suggest this indicates some sort of further read quality or map quality filtering scheme should be applied for METEORE to prevent this.

12. It's somewhat surprising that nanopolish and METEORE (RF) (which uses megalodon and deepsignal) report higher methylation at the CGI for TPO than the other tools. Why don't megalodon or deepsignal report higher methylation - I thought they were driving the METEORE results heavily. A zoom in of individual methylation calls in this region might be informative. A cursory examination suggests this could be allele-specific methylation in this region (0.5 methylation frequency).

13. The authors should also plot WGBS coverage in these regions - WGBS is far more subject to variable coverage than the nanopore sequencing because of the resulting GC bias, especially (I believe) the older WGBS dataset they are drawing from.

14. Small point - I wonder about the bandwidth of the loess being used in Figure 5, since the curves look . . . pretty smooth. I'd be interested in seeing the points.

ANSWERS TO REVIEWERS' COMMENTS (in blue)

Reviewer #1 (Expertise: Nanopore technologies for DNA sequencing/DNA modification prediction):

In this study, the authors present a systematic benchmarking of six tools for the detection of CpG methylation using Nanopore sequencing. Based on the evaluation, they also proposed a consensus approach (METEORE) to improve accuracy over individual tools. The benchmarking includes evaluation using individual reads, control mixtures of methylated and unmethylated reads, and Cas9-targeted sequencing and whole-genome bisulfite sequencing. The evaluation of the tools can help understand and improve the prediction of CpG methylation using Nanopore reads. I have two major concerns:

Major Concerns:

1. The performance of METEORE, which combined DeepSignal and Megaloddon, did not improve much comparing with those of original DeepSignal and Megaloddon tools.

We agree that this is true for Megaloddon. However, Megaloddon is much slower than the other methods on a CPU. In contrast, Nanopolish and Tombo are the fastest 5mC callers on a CPU, but their individual accuracy is low.

Nanopolish is very popular because it only requires the fastq file after basecalling, which is readily produced by the Minknow software. This ease-of-use is contrasted with a low accuracy, as we have shown in the manuscript. The advantage of METEORE is that this low accuracy can be improved when combined with a second method. METEORE also provides snakemake pipelines to make it easy to run all the methods.

Additionally, for methylation mixes, the combination with METEORE (DeepSignal + Megaloddon) had the lowest proportion of sites outside the 10% window (Fig. 3f), and much lower than DeepSignal or Megaloddon individually. This is important, as intermediate methylation levels are hard to predict in general.

In our manuscript, we argue that METEORE provides a way to combine fast CPU-based methods to reach an accuracy comparable to Megaloddon. We have tried to make this point more clear in the discussion section.

2. The evaluation of the correlation of CpG prediction using WGBS data and different coverages of Nanopore reads did not make sense. What the authors used is of the different levels of coverages, but the coverage > minimum coverage. Figure 4, supplemental Figure 9 and 10 using "Nanopore coverage" as X-axis is confusing. With more coverage of Nanopore reads, the correlation of CpG prediction using WGBS and Nanopore reads should increase. The performance decreasing in Figure 4, supplemental Figure 9, and 10 are due to the fewer CpG

reads covered by high coverage reads. To demonstrate the correlation of CpG prediction using WGSB and different coverage of Nanopore reads, the authors should sample different coverage of Nanopore reads and make the prediction, as did in methyltransferase paper.

In our original analysis, we tried to provide useful information about how to select the sites based on the measured coverage in an experiment. We agree that the suggested analysis is better to obtain a balanced estimate of the accuracy in terms of coverage. We thus performed the suggested analysis using the data from Gilpatrick et al. We subsampled reads to build datasets with coverage 5x, 10x, 20x, 50x for the CpG sites. As suggested by the reviewer, the correlation improves with coverage. The results of this analysis are now shown in Supp. Table 8:

Supp. Table. 8. Pearson correlation values of methylation frequencies obtained from Nanopore reads and bisulfite sequencing under different read coverages.

	5x	10x	20x	50x
Nanopolish	0.7307	0.7719	0.7828	0.7905
DeepSignal	0.7851	0.8149	0.8206	0.8294
Megalodon	0.7999	0.8226	0.8292	0.8371
Guppy	0.6757	0.7161	0.7432	0.7545
Tombo	0.6281	0.6655	0.6892	0.7028

Furthermore, to improve the clarity of our figures, the labels for the x-axis for Fig. 4b, 4c, Supp. Fig. 9b, 9c and 10 were changed to “Minimum nanopore input coverage” (see also answer to reviewer 3’s point 10).

some minor comments:

1. On page 4 “p-value < 2.2.e-16 for all tools”, how were the p-values calculated?

We used the `cor.test()` function in R which outputs both the correlation coefficient and the p-value (significance level) of the correlation. The p-value is calculated using the t-test for N-2 degrees of freedom, where N represents the number of points. This is the level of resolution of the test. We are indicating that the p-value is smaller than that.

`cor.test()` is part of the stats package from R, so we cited the R project in the paper: R Core Team (2020). R: A language and environment for statistical computing. R Foundation for Statistical Computing, Vienna, Austria. URL <https://www.R-project.org/>.

2. On page 4, “Supp. Table 2” was quoted 2 times, and it seems that either the two quotes are the same (there is no need to quote/repeat the sentence twice), or one of them is not quoted appropriately.

We have deleted one of the sentences.

3. On page 17, “bi-sulfite” should be “bisulfite”, to be consistent with the whole manuscript.

We have corrected this typo.

4. On page 18, the methylation LLR threshold of Nanopolish has changed from 2.5 to 2.0 since v0.12.0. The authors should check the actual threshold used in their analysis.

We repeated the analysis of the mixture dataset 1 with Nanopolish using the 2.0 threshold. There are almost no differences in the results in comparison with the 2.5 threshold (see new Supp. Fig. 19). In particular, the high dispersion and high false positive rate observed before remains present with the 2.0 threshold. Accordingly, a change to 2.0 would not affect our conclusions. For this reason, and to keep it consistent with earlier published analyses, we have maintained the 2.5 threshold. We have indicated this explicitly in the Methods section.

We have indicated in Methods that the default threshold has changed, and that we did not see major differences between both thresholds.

5. The analysis results of Megalodon in Supp. Fig. 8b and Supp. Table 7 are not the same. The authors should check which one is right.

Supp. Table 7 is the correct one. We have updated the results of Megalodon in Supp. Fig 8b.

6. How did the N in Table 1 obtain?

N is the number of observations in x and y variables and they were obtained from the correlation test (`cor.test()` function in R), where $N = \text{degrees of freedom} + 2$.

We have added a reference from R in the method section.

Reviewer #2 (Expertise: Nanopore technologies for DNA sequencing/DNA modification prediction):

I have no comments on the paper. Its a nice study on comparison of tools for detection of methylation using nanopore sequencing data. The findings are reasonable and I believe would be useful to sequencing centers and researchers in this area.

We thank the reviewer for the positive and supportive comments.

Reviewer #3 (Expertise: Nanopore technologies for DNA sequencing/DNA modification prediction):

In this manuscript, the authors set out to benchmark software for calling 5-methylcytosine (5mC) base modifications from Nanopore data. They compare six state-of-the-art software tools, and, unsatisfied with the accuracy provided, created their own 5mC calling pipeline (METEORE) by combining two or more of these six tools. Then they went on to evaluate this against the other tools in both the *E. coli* genome and targeted locations in the human genome.

Notably, the authors:

- A. Created a pipeline that standardizes inputs and outputs so that different tools can be easily compared. This also helps METEORE to be more effectively used in the future.
- B. Created a clever test set for their bakeoff analyses but using publicly available “100% methylated” reads to create different methylation proportions.
- C. Performed comparisons between the tools including how different methylation calling cutoffs affect tool performance.
- D. Applying their tool and their analyses to human data.

We appreciate the reviewer's positive evaluation of the value of our work and the helpful comments to improve the analysis and presentation of the data.

However, I still have some significant problems with the analysis:

1. Small point - megalodon is also a neural net based model - after anchoring the data to the reference, the tool uses the neural net Guppy output and viterbi decoding to find the best path to classify modifications (if I'm not mistaken)

We have added this in the Methods section in the paragraph about Megalodon.

2. Figure 2 was . . . hard to parse, I think because of the number of different things being plotted on the same graph. 2A for example it is easy to see the dispersion (of nanopore for example), but hard to see the linearity. I'd consider breaking this into separate plots for clarity, or putting some of it in the supplement to make the most critical points. A box-plot instead of violin may allow more clarity.

We consider that violin plots are more adequate to show distributions than boxplots in this case, as they capture better the density of points along the distribution. Fig. 2a shows in a succinct way several relevant features in the prediction of methylation. One of them is the high dispersion shown by most of the tools, which has been widely underappreciated in the context of methylation prediction with Nanopore reads.

For improved clarity, we have included an additional plot (Supp. Fig. 2) only showing the correlation lines.

2b-d I'd suggest some sort of ecdf plot or other line based plot instead of bars - the bars just take up space when you could have lines to show the data just as well.

As Figures 2c and 2d depict the proportions of true negatives and false positives, respectively, for an increasing value of a threshold (methylation frequency), we agree that these are better represented as Empirical (or Estimator) of the Cumulative Distribution Function (ECDF) plots. The data from Figures 2c and 2d are now represented as ECDF plots, as suggested:

We did not show an ECDF plot for Figure 2b as this data is not related to a changing threshold, i.e. there is not a changing range of values defined by a sliding cut-off to use for the cumulative distribution. It is rather showing a percentage in each individual mixture dataset.

3. Are the authors using map quality scores for filtering the aligned reads - this is a stringent cutoff in some tools, but not in others and could easily affect the quality of the resulting calls.

We are not imposing any additional filtering cutoff. All the tools used for benchmarking are using minimap2 as default aligner and we did not change the default setting for the tools. Although the discrepancy in the filtering criteria by the tools may affect the quality of the methylation call, we considered that each tool should address this in their underlying algorithm. For example, Tombo does not perform any filtering but does the requiggle with the raw signals and only considers the best matched reads. We run the tools with the default setting exactly as the typical user will do.

This is how each tool processes the read data:

- Nanopolish uses a mapping quality ≥ 20
- Tombo does not filter out any reads, but it performs a re-squigglng.
- DeepSignal uses the re-squigglng algorithm from Tombo
- Megalodon has no filter
- Guppy filters based on the read alignment. From the documentation: “*The minimum read coverage required to consider alignment a success. No alignment results below this threshold will be output. The default value is 0.6. If the aligner reports more than one possible alignment, only the best one is output. An alignment that covers less than 60% of the read or of the reference will be rejected.*”
- DeepMod uses a mapping quality > 10

4. Choice of CpGs for analysis:

a. The majority of the benchmarking in the manuscript is based on 100 “random” CpGs from the E. coli genome. As the authors are no doubt aware, nanopore sequencing derives its signal

from a k-mer based model, with (presumably) some k-mers performing better than others. The size of the signal window is even different between the different tools. This can result in some tools performing better and worse in different areas. This is exemplified by the fact that the authors found different numbers on the second set of 50 CpGs, but the k-mer contexts of these CpGs are not clear (comparing Figure 2A to Figure 3E).

To study how sequence context might influence the prediction of methylation in datasets 1 and 2, we first built sequence logos for each of these two sets of CG sites. Both datasets show similar sequence biases around the CG sites:

These SeqLogos show the probability of each base (left panels) and the information content (right panels) for each position around the selected CGs, and have been added in the new Supplementary Figure 1.

Although there is a difference in performance between the datasets 1 and 2, not all methods show decrease in performance for dataset 2 with respect to dataset 1. DeepSignal, which uses 17-mers, shows a decrease in performance. Tombo and Nanopolish use 5-mers, and whereas Tombo has a slightly higher correlation in dataset 2, Nanopolish shows a marginal decrease. In contrast, Megalodon does not show differences between Datasets 1 and 2. To further investigate how sequence may affect the prediction accuracy, we have carried out a thorough analysis that is described under the points 4c and 4f below.

b. While some of these differences are small, the authors use small differences to claim that their tool outperforms all other tools (“METEORE RF combining Megalodon and DeepSignal achieved lower RMSE compared with the individual tools (Fig. 3e)”; page 9). For example the difference between DeepSignal for mixture 1 and mixture 2 is larger than the difference between METEORE RF and Megalodon in mixture 2. It is not clear to me why the authors can’t analyze the entire singleton set of the E. Coli genome?

Dataset 2 is used to perform an independent validation of the best combination of tools, which is estimated (i.e. “learned”) from dataset 1. The use of an independent dataset is needed here to validate the hypothesis that a specific combination of two tools improve over the individual tools.

Dataset 2 is also used as an independent validation set for testing the new scores cut-offs (different from the default ones) that were estimated from the dataset 1.

Megalodon shows a consistent accuracy across both sets, whereas other tools like Nanopolish and DeepSignal are not as consistent. However, the relevant comparison in relation to the accuracies of the methods is within each dataset rather than across datasets. In this regard, we show that using METEORE, DeepSignal accuracy can be improved when combined with a second tool. In fact, except for Megalodon, which already shows a high accuracy in both datasets, a combination of two methods improves over individual methods in general.

We also considered the possibility that the M.SssI treatment is not 100% effective, hence it may cause the observed variability. The analyses below (see answers for point 4c below) show that the prediction methods present high variability even at sites where methylation treatment (positive cases), or lack of methylation treatment (negative cases), is most efficient.

c. The CGs discussed are also profiled by Illumina bisulfite sequencing data in Simpson et al, with that data provided in ENA, the authors should consider examining the exact bisulfite provided methylation levels rather than estimating since they are looking at specific CGs.

To clarify the reviewer’s comment, in the paper we did not estimate or simulate the methylation levels. The control mixtures (datasets 1 and 2) were built using nanopore reads from two different sample controls, one unmethylated (PCR) and one that had been treated with M.SssI (PCR+M.SssI) (methylated) from Simpson et al. Therefore, in the paper we are not estimating the methylation level. Instead, we are using the individual reads from the two methylation controls to generate a ground truth for testing the tools.

The unmethylated sample has a conversion rate of ~0.26% in the sites from the datasets 1 and 2. The methylated sample shows an average conversion rate of >94% in both datasets. This suggests that not all sites may have 100% efficiency for M.SssI treatment. To test whether this

may explain the dispersion shown by the methods for dataset 1 and 2, we performed an additional experiment. We used reads from the unmethylated sample and considered only those sites that showed exactly 0% methylation in the bisulfite sequencing (in the same sample). These were considered to be the most reliable sites for 0% methylation. Similarly, we used reads from the methylated sample and only considered the sites that showed exactly 100% methylation in the bisulfite sequencing (in the same sample). These were considered to be the most reliable sites for 100% methylation. We calculated these two sets of sites for dataset 1 and dataset 2. For each one, we thus obtained two sets of reads and sites: one set of unmethylated reads and the most reliable sites for 0% methylation, and one set of methylated reads and the most reliable sites for 100% methylation. We then plotted the methylation frequency predicted by each method from these reads:

We found that the tested methods showed a similar behaviour as in our analysis using all sites: Nanopolish and Tombo showed a high number of false positives but high recall for fully

methyated sites, whereas Guppy and Megalodon had low false positive rates but high false negative rates; and all methods showed a high dispersion. This analysis has been added to the revised manuscript, and the plots above are now shown in Supp. Fig. 18.

d. Failing this, or even in preference to it, Simpson et al also contains a PCR amplified and M.SssI treated GM12878 control set which could provide another control set with even more k-mer contexts. This might be useful to demonstrate the performance on a human genome - the putative end goal. I know the authors used the targeted human data later to evaluate, but this is not the same as a more agnostic evaluation (more on the targeted data later)

We have added a comprehensive analysis of k-mers and G+C content in relation to accuracy for each method in our answer to point 4f below.

e. The authors specifically test the performance of their tool on singleton CGs, i.e. CGs with no other CG within 10 bases up or downstream. This is fair to eliminate signal from other nearby CGs, but presumably is a very different evaluation than operation on a human genome, with relatively dense CGs in CGI (for example).

For the control mixture datasets (1 and 2) we analysed singleton CpG sites. However, for the comparison with bisulfite sequencing using nCATS, we used all the CpG sites regardless of context.

Below, next to each graph shown in Fig. 5 with a CpG island (CGI), we show the LOESS smoothing line plots of the CGI region, and the SeqLogo for the sequence context of the CpG sites:

These plots have been added as Supplementary Figure 15.

f. This all fits into the point: Identifying k-mers where a tool performs better or worse would be very useful.

To address this question, we calculated for each method the absolute value of the difference between the prediction made from the Nanopore reads and the WGBS for each CpG site, i.e. $|\text{Nanopore} - \text{WGBS}|$. In the plot below we show the mean absolute difference for each method (y-axis) for all different 8-mer sequences with a CpG in the center (XXXCGXXX) (x-axis), ranked from left-to-right in ascending order according to the average value across methods.

This plot clearly indicates that there is a subset of CG-containing 8-mers in which all methods agreed with the WGBS (left side of the x axis), and a different subset where all methods most disagreed with WGBS (right side of the x axis). We selected the top and bottom 40 8-mers according to the average discrepancy across methods and found that Nanopore and WGBS disagreed the most at AT-rich k-mers (see the sequence logo below).

This figure shows in the form of two SeqLogo's, the sequence biases associated to the “bad” sites (upper panel) and “good” sites (lower panel), and indicates the sites that have a significant enrichment in any of the two sets with respect to the other one (read line). This was performed with the tool pLogo (O'Shea et al., 2013). The statistical comparison indicated that Nanopore and WGBS tend to disagree most at CG sites that occur in AT-rich 8-mer context (i.e. XXXCGXXX).

To investigate the specific biases for each tool, we categorized the k-mers into “good” or “bad” for each tool based on their specific distributions of absolute values. We defined as good sites those with absolute difference of 0, whereas sites with absolute difference > 0.5 were considered bad.

In the right panel of the figure, we show in the form of two SeqLogo's, the sequence biases associated to the bad sites (upper panel) and good sites (lower panel). The statistical comparison showed no significant biases in relation to the discrepancy with WGBS. However, the test indicated that Nanopolish predictions agree most with WGBS when there is a G preceding the CG, i.e. XXGCGXXX.

A similar comparison for DeepSignal showed that bad predictions are significantly associated with a T at position 1 of the 8-mers and with an A at position 8 of the 8-mer.

For Megalodon, no significant association was found, although the logos indicate a possible association of an A at position 8 of the 8-mer with bad prediction, similarly to DeepSignal.

Tombo showed a clear association of T+A rich sequences with bad predictions, and a clear bias for 8-mers of the form XXGCGCXX for good predictions.

Guppy showed a significant association with T at position 1 and with G at position 6 for bad predictions, and an association to XXXCGCXX for good predictions.

DeepMod showed a strong bias for T+A rich sequences in bad predictions, with significant enrichment at several positions.

METEORE-RF and METEORE-REG, as used in the manuscript, show no significant biases. Only a trend of bad predictions in 8-mers with an A in the position 8, similar to Megalodon.

These plogo plots have been added as Supplementary Figure 17.

To further assess how the accuracy depends on the nucleotide content of the CG context, we separated all 8-mers into four groups according to the percentage of C or G (%C+G). We then plotted the distributions of the absolute differences, comparing the Nanopore prediction with the WGBS value, for each tool in each of these 8-mer groups:

This plot indicates that for most of the tools, there are no strong biases with C+G content, except for Guppy and DeepMod, which we already observed in the analyses above. This plot has been included as Supp. Fig. 16.

We performed a similar analysis using the actual sign of the difference (Nanopore - WGBS) rather than the absolute value. In this way, a positive value indicates the Nanopore methylation tool overpredict and a negative value indicates the tool underpredict compared with WGBS. Overall, the results were very similar to those reported above. We ranked the k-mers according

to the average differences across methods:

Overall there were more 8-mers with overpredictions, i.e. difference of Nanopore - WGBS > 0.5 , than with underprediction, i.e. difference of Nanopore - WGBS < -0.5 . The exception was Guppy, which had more 8-mers with underpredictions than overpredictions.

Again, we identified statistically significant k-mers using pLogo (O'Shea et al., 2013). However, in some cases there were not enough occurrences of overprediction/underprediction for some tools to perform the test, so the sequence logos are not available in these cases.

Nanopolish showed no significant biases comparing overprediction or underprediction vs good 8-mers.

DeepSignal showed a significant association with an A at position 8 for overpredictions. However, there were only 4 8-mers for underpredictions (difference between -0.5 and -1), so we could not perform the analysis.

pLogo statistics and parameters

foreground (fg)	overprediction_mgld
foreground size	11
background (bg)	good_prediction_mgld
background size	253
pLogo width	8

statistically significant positions

position	frequency	value	fixed

Overpredictions for Megalodon showed no significant biases, but the same high frequency of A at position 8 as before. There were only 2 8-mers of underprediction (between -0.5 and -1) so we could not perform the analysis.

pLogo statistics and parameters

foreground (fg)	overprediction_tb
foreground size	102
background (bg)	good_prediction_tb
background size	143
pLogo width	8

statistically significant positions

position	frequency	value	fixed
A at 1	33.33%	2.85307	[ ]

pLogo statistics and parameters

foreground (fg)	underprediction_tb
foreground size	7
background (bg)	good_prediction_tb
background size	143
pLogo width	8

statistically significant positions

position	frequency	value	fixed

Tombo overpredictions showed the same biases found before. For underpredictions no significant biases were found.

Guppy underpredictions showed a significant association with a G at position 6 and with a T at position 1. There were only two 8-mers with overprediction (>0.5 difference), so we could not make the statistical assessment.

As before, METEORE (RF) and METEORE (REG) showed no biases. The underpredictions did not have enough 8-mers to perform the calculation.

As before, we plotted the differences Nanopore - WGBS for four different 8-mer bins according to %C+G:

There was a greater variability for DeepMod and Guppy across the bins, with a strong bias towards under prediction. Although there seemed to be more outliers for all tools at higher %C+G, the distributions appeared to be also narrower.

Overall, we found that sequence biases associated with worse predictions are generally related to the overpredictions in comparison with WGBS, except for Guppy, which are linked to underprediction. Overall, most methods tend to overpredict methylation with respect to WGBS, and the majority of methods (DeepMod, Tombo, Guppy, DeepSignal) show some sequence bias.

5. The authors have developed a combination of tools to call methylation - trying to improve on the weaknesses of any given tool with either linear regression or random forest on the outputs. Ideas like this already exist for structural variants (e.g. Parliament), but in theory this could provide improved accuracy in methylation calling. But the authors do not assess concerns of runtime/resources; many of the tools the authors list already have substantial runtime/hardware requirements - running more than one provides an additional burden. And seemingly for a small incremental improvement over megalodon alone. **The authors need to make a clear point that adding another tool (even just deep signal) is “worth” the effort, because that isn’t clear from the current results.** This should especially be benchmarked on larger genomes - though the targeted sequencing is certainly a valuable application, whole genome methylation will also likely be an important application.

We have expanded on this point in the discussion. The combination of Megalodon and DeepSignal presents a good balance for accuracy and running time in the case of GPUs. Without GPU support, the combination of Nanopolish and DeepSignal can achieve accuracy similar to Megalodon in a CPU-only environment, with good running times. This is based on our analyses provided in Supp. Table 13.

6. I am confused by the optimum cutoff settings, especially in terms of Megalodon (the tool with the closest performance to METEORE). While the cutoffs improved general metrics of methylation calling, when you compare methylation percentages outside of the windows in Fig.

3f to Supplemental Fig. 3c/3d, the optimized cutoffs for many of these tools made the tools `_worse_` than default settings. Can you explain how an “optimized” setting would make the calls worse? It would be a worthwhile point to explain this in the paper as well.

We have tried to clarify this in the text. We hypothesized that it would be possible to identify cutoffs for the tools, different from the default ones, that could improve their accuracy. The first strategy used was based on the score distributions and accuracy metrics on dataset 1. We identified the single score cutoff that would maximize TPR - FPR, or minimize $FPR^2 + (1 - TPR)^2$. The word “optimization” was used in the sense of finding this maxima or minima. In practice, this approach did not cause an improvement in some of the tools, due to how scores were distributed between positive and negative cases. The second strategy used was to identify a double cutoff and remove all reads with scores in between these two selected values. Here the aim was similar: trying to identify alternative score cutoffs that would improve the accuracy of the tools. We have rephrased the text to make it clearer.

7. It is not 100% clear, but it seems in the bisulfite work that the authors are no longer requiring that the CGs be distant from each other? Given that it is more relevant to plot local CG density rather than just GC% per se.

That’s correct. In the comparison with the bisulfite data, we are using all CG sites. We have compared tools across different % C+G as suggested (see our answers to the point 4f). We have also added in the Supplementary the plots of the comparisons with the bisulfite data indicating the CG sites on those regions.

8. I like the comparison of METEORE to WGBS. However, I again believe that the authors should expand the analysis on the data at hand and dig into `_where_` (i.e. what k-mers) have differences beyond just correlation and broad patterns at promoters. This would likely be very informative for those in the community going forward to understand the limitations of these tools. In this vein, it would be worthwhile to better visualize WGBS methylation percentages in Figures 5 and Supplemental 11/12. It is difficult to see (grey lines) and that is probably the most relevant comparison in those figures.

We have changed the grey line to black line for WGBS in Fig. 5 and Supplementary Figs.13, 14, and 15, for better visualization. Please see our answers to point 4f where we have answered the question about the k-mer contexts and GC biases.

9. I’d be more interested in a faceted scatter plot of methylation versus bisulfite than the overall “binned” methylation shown in Figure 4a - I think the bins lose subtlety.

We already had the scatter plots comparing the methylation detected by each tool with bisulfite (previous Supp. Figs. 6 and 8). We have now added the correlation bands to those plots as suggested (now in Supp. Figures 8 and 10).

10. On 4b: It seems like using the “filtered” read coverage for nanopolish and tombo provides a bit of a skewed picture of the coverage required to achieve correlation, since in reality you need

a higher coverage (~60X) to get the 35X nanopolish coverage reported? I would suggest reporting the coverage of data `_input_` into nanopolish instead.

For Fig 4b, 4c, Supp. Fig. 9b, 9c, 10a and 10b, we used the coverage data output reported by each tool. Some methods discarded sites in reads and so the resulting coverage reported by the tool was less than the corresponding coverage of Nanopore reads. We agree that we should report the input coverage calculated from the raw coverage per base from the input BAM file (x axis) and plot the correlation (y axis) as a function of it for Fig. 4b, Supp. Fig. 9b and 10a. Instead of plotting the number of sites in log2 scale on the y axis on Fig. 4c, Supp. Fig. 9c and 10b, we present the mean coverage reported by each tested tool against the minimum input coverage. All these plots have been updated in the manuscript and supp. material.

11. More on 4b - it seems that there is a big dip in correlation of both METEORE modes at 55X - and generally that higher coverage gives worse results? The authors do not offer an explanation for this, but I'd suggest this indicates some sort of further read quality or map quality filtering scheme should be applied for METEORE to prevent this.

The drop in accuracy at high coverage thresholds (minimum coverage requirements) is due to the reduced number of sites available. We have added a sentence in the text related to that figure. We have also performed an new analysis of the Pearson correlations as a function of the coverage maintaining the number of sites constant (see Reviewer 1's major point 2). These new results show that the correlations improve with increased coverage.

12. It's somewhat surprising that nanopolish and METEORE (RF) (which uses megalodon and deepsignal) report higher methylation at the CGI for TPO than the other tools. Why don't megalodon or deepsignal report higher methylation - I thought they were driving the METEORE results heavily. A zoom in of individual methylation calls in this region might be informative. A cursory examination suggests this could be allele-specific methylation in this region (0.5 methylation frequency).

The higher values of Tombo, Nanopolish, and METEORE (RF) are likely due to the way their particular models handle high dispersion of methylation scores. METEORE (RF) consistently gives higher methylation call values than most other callers (see Supp. Fig. 13 and Supp. Fig. 14). This is likely a combined result of the training data used, plus the number of available parameters and depth of the random forest model. Below we show a zoom in of individual methylation calls at the CGI for TPO region.

We observe many outliers with WGBS methylation scores, but these appear evenly distributed above and below the LOESS fit line. This high dispersion will force down the average methylation frequency reported by WGBS.

13. The authors should also plot WGBS coverage in these regions - WGBS is far more subject to variable coverage than the nanopore sequencing because of the resulting GC bias, especially (I believe) the older WGBS dataset they are drawing from.

We used WGBS data from ENCODE (GM12878). This is the coverage for our nCATS regions we used in our analyses:

This is the coverage for the nCATS regions from Gilpatrick et al. (2020) used in our analyses:

14. Small point - I wonder about the bandwidth of the loess being used in Figure 5, since the curves look . . . pretty smooth. I'd be interested in seeing the points.

We used the `geom_smooth()` function from `ggplot2` that uses a default bandwidth of 80 points, and which was also used by Gilpatrick et al. 2020 (see script `191006_plot_Methylation_flg_min_ill.R` in <https://github.com/timplab/Cas9Enrichment>). Smoothing using the default bandwidth value also provides a clear description to the overall change of methylation along the regions. However, we are showing the points, as requested by the reviewer, in the plots provided in answer to 4e and 12 above (included in Supp. Fig 15).

Reviewers' Comments:

Reviewer #1:

Remarks to the Author:

I would like to thank the authors for their detailed responses to my comments. The manuscript reads much better and is more clear than before. I only have some minor comments.

1. In supplemental table 8, please also add the performance of METEORE under different levels of coverage.
2. Since METEORE is a combination of DeepSignal and Megalodon, it is unclear how METEORE uses less CPU and Memory than both in supplemental Table 13.

Reviewer #3:

Remarks to the Author:

The authors have fully addressed all of my major comments and I now support publishing this manuscript. The k-mer specific analysis is an especially valuable addition. The results are very interesting and we believe that they will be valuable to the community. The way the authors expanded upon some of the finer points of the analyses (CPU vs. GPU run time) was also nice.

A few minor points:

1. I would be interested in seeing the underlying data for the kmer analysis, especially for Supplemental Figure 16A (8-mers ordered by average dispersion). Can you include that data as a table in the Supplementary Data?
2. Page 10 - "Combining the methylation predictions from both stands on CpG sites showed an improved correlation for all tools compared with using the methylation prediction independently for each strand (Supplementary Fig. 8). Using the combined methylation from both stands, all tools showed a positive correlation of the with WGBS signals (Table 1)." - Make sure "strand" is used throughout. It seems like this is the only spot of this error.
3. Page 23 - Change "Bi-sulfite" to "bisulfite" - Double check all of these. It seems like this is the only remaining one.

REVIEWERS' COMMENTS

Reviewer #1 (Remarks to the Author):

I would like to thank the authors for their detailed responses to my comments. The manuscript reads much better and is more clear than before. I only have some minor comments.

1. In supplemental table 8, please also add the performance of METEORE under different levels of coverage.

We have added the performance of METEORE (RF) and METEORE (REG) to the Supplementary Table 8.

2. Since METEORE is a combination of DeepSignal and Megalodon, it is unclear how METEORE uses less CPU and Memory than both in supplemental Table 13.

The CPU time and memory in Supplementary Table 13 is only the run of Meteore on two data files. We have clarified in the legend: "For METEORE, we only recorded the time needed for running a single Python script to make the per-read consensus predictions, which was independent of the two methods being combined. Here we used METEORE combining DeepSignal and Megalodon using a random forest (RF) (parameters: max_depth=3 and n_estimator=10) and a regression (REG) model, where both models took per-read prediction outputs generated from the selected tools. To estimate the overall CPU time and memory used, any two tools can be considered, plus the overhead of an additional step of METEORE."

Reviewer #3 (Remarks to the Author):

The authors have fully addressed all of my major comments and I now support publishing this manuscript. The k-mer specific analysis is an especially valuable addition. The results are very interesting and we believe that they will be valuable to the community. The way the authors expanded upon some of the finer points of the analyses (CPU vs. GPU run time) was also nice.

A few minor points:

1. I would be interested in seeing the underlying data for the kmer analysis, especially for Supplemental Figure 16A (8-mers ordered by average dispersion). Can you include that data as a table in the Supplementary Data?

It is provided in Supplementary Data 13.

2. Page 10 - "Combining the methylation predictions from both stands on CpG sites showed an improved correlation for all tools compared with using the methylation prediction independently for each strand (Supplementary Fig. 8). Using the combined methylation from both stands, all tools showed a positive correlation of the with WGBS signals (Table 1)." - Make sure "strand" is used throughout. It seems like this is the only spot of this error.

Thanks for pointing this out. We have corrected them.

3. Page 23 - Change "Bi-sulfite" to "bisulfite" - Double check all of these. It seems like this is the only remaining one.

Thanks for pointing this out. We have corrected it.